# Caecal microbiota compositions from 7-day-old chicks reared in high-performance and low-performance industrial farms and systematic culturomics to select strains with anti-*Campylobacter* activity

**Aurore Duquenoy[1], Maryne Ania[1], Noémie Boucher[1], Frédéric Reynier[1], Lilia Boucinha[1], Christine Andreoni[2], Vincent Thomas[1]***

**1** Bioaster, Paris, France, **2** Boehringer Ingelheim Animal Health, Lyon, France

\* vincent.thomas@bioaster.org

**Data Availability Statement:** Partial 16S-rRNA encoding gene sequences of cultivated isolates

## Abstract

There is growing interest in exploring the chickens' intestinal microbiota and understanding its interactions with the host. The objective is to optimize this parameter in order to increase the productivity of farm animals. With the goal to isolate candidate probiotic strains, specific culturomic methods were used in our study to culture commensal bacteria from 7-days old chicks raised in two farms presenting long history of high performance. A total of 347 isolates were cultured, corresponding to at least 64 species. Among the isolates affiliated to the Firmicutes, 26 had less than 97% identity of their partial 16S sequence with that of the closest described species, while one presented less than 93% identity, thus revealing a significant potential for new species in this ecosystem. In parallel, and in order to better understand the differences between the microbiota of high-performing and low-performing animals, caecal contents of animals collected from these two farms and from a third farm with long history of low performance were collected and sequenced. This compositional analysis revealed an enrichment of *Faecalibacterium*-and *Campylobacter*-related sequences in lower-performing animals whereas there was a higher abundance of enterobacteria-related sequences in high-performing animals. We then investigated antibiosis activity against *C. jejuni* ATCC 700819 and *C. jejuni* field isolate as a first phenotypic trait to select probiotic candidates. Antibiosis was found to be limited to a few strains, including several lactic acid bacteria, a strain of *Bacillus horneckiae* and a strain of *Escherichia coli*. The antagonist activity depended on test conditions that mimicked the evolution of the intestinal environment of the chicken during its lifetime, i.e. temperature (37˚C or 42˚C) and oxygen levels (aerobic or anaerobic conditions). This should be taken into account according to the stage of development of the animal at which administration of the active strain is envisaged.

were deposited in GenBank under accession numbers MN913773-MN913871 (see S3 Table). 16S rRNA gene repertoires of caeca collected from high- and low-performing farms were deposited in SRA under accession number PRJNA600473. The genome of the C. jejuni field isolate has been deposited under GenBank accession number JACBXE000000000.

**Funding:** This work has received, through BIOASTER investment, funding from the French Government through the Investissement d'Avenir program (Grant No. ANR-10-AIRT-03). This study was partially funded by Boehringer Ingelheim Animal Health. Dr Christine Andreoni, who is employed by Boehringer Ingelheim Animal Health, was involved in study design, decision to publish, and preparation of the manuscript.

**Competing interests:** This study received partial funding from the company Boehringer Ingelheim Animal Health, by which the co-author Christine ANDREONI was employed at time of the study. This does not alter our adherence to PLOS ONE policies on sharing data and materials.

## Introduction

Among other food-producing animals markets, the huge, constantly increasing chicken meat market represents a major economical stake, with approx. 25 billion of live animals representing 122 million tons (carcass weight equivalent) of meat output in 2017, and a 21.3 million tons increase in world poultry meat production between 2010 and 2017 (http://www.fao.org/faostat). Antibiotics at subtherapeutic levels have historically been used as growth promoting agents in this industry [1]. These treatments likely have an impact on chicken microbiota composition [2, 3] but the exact mechanisms that promote animal growth are not completely explained.

Nowadays, the use of antibiotics as growth-promoting agents are being banned in certain regions to prevent selection of antibiotic resistant bacteria [4]. This antibiotic ban stressed the need for alternative means of improving critical parameters such as resistance of poultry to infections and feed efficiency, which represents the amount of feed required (in kg) to produce 1 kg of poultry meat. In this context, the chicken gastrointestinal microbiota has emerged as an important parameter that should be taken into account to increase animals' performances. It can potentially improve nutrition efficacy as well as resistance to infections or colonization by zoonotic infectious agents [5, 6]. However, its composition varies according to a number of parameters including age, nutrition, seasons [7, 8] but also the rearing environment and specific process such as litter management regimen that can potentially impact productivity by contributing to poultry gut microbiota composition [9–11]. The impact of the environment on chicken microbiota composition is increased by the fact that the eggs are separated from the laying birds, and then incubated in clean conditions, with the newly hatched chicks being consequently very susceptible to colonization by environmental bacteria. Consistent with these observations, adult chickens from the same batches of chicks but subsequently reared on different farms may have different intestinal microbiota and different performance indexes. Consequently, it can be hypothesized that modulating poultry microbiota composition towards gut microbiota compositions observed in high-performing farms could be an efficient strategy to improve overall productivity. In line with this hypothesis, Fecal Microbial Transplant (FMT) experiments have been reported to slightly improve growth performance in female chickens although not dramatically impacting chicken's gut microbiota composition [12]. Other possible interventions include inoculation of newly hatched chicks with adult caecal content that was expanded in chemostats [13], or spraying caecal contents directly onto eggs to colonize the embryo through the eggshell [14]. However these approaches are still relatively experimental and the use of pre- as well as pro-biotics represent a more practical option. Probiotics have been used for years in poultry production, with (for some of them) documented beneficial effect on productivity [4]. Biological mechanisms underlying this positive effect are not always well understood but could be linked to improved digestion of nutrients, improved resistance to colonization by various pathogenic species (barrier effect), improved gut mucosa barrier function and/or improved immune-stimulation [15]. To date, most probiotic strains have been limited to lactic acid bacteria (LAB), *Bacillus* species and yeasts. The detailed compositional analysis of chicken microbiota, made possible by next-generation sequencing technologies, can potentially provide a better understanding of the differences between high-performing and low-performing chicken microbiota. This understanding should result in a better ability to identify new probiotic candidate species, including strict anaerobic commensal, spore-forming or non-spore-forming species. An important criterion in the selection of these candidate strains is their ability to confer resistance to infection by pathogenic microorganisms and/or colonization by zoonotic commensals, which can be due to different mechanisms including the production of antagonistic compounds [16] or

competition for specific substrates [17]. Among other antagonistic activities, the ability of probiotics to confer resistance to colonization by *Campylobacter jejuni* is often studied due to the zoonotic potential of this commensal species, but also due to an association with lower productivity of colonized chickens [18]. Colonization of chickens by *C. jejuni* is usually reported around days 15–20 of the chicken life cycle but has also been reported in younger chicks, especially in free-range birds for which environmental exposure can potentially be high [19, 20]. Short chain fatty acids (SCFA) produced by the commensal microbiota seem to play an important role in *C. jejuni* capacity to colonize animals [21]. Specific butyrate-sensing mechanisms have been recently described, altering transcription of target genes and being therefore proposed as a vital mechanism allowing *C. jejuni* to recognize and colonize specific intestinal niches [22, 23].

In this work, several culture conditions were used to isolate and culture a variety of bacterial species from chicks raised in high-performance conventional farms, with the goal to build a collection of commensal strains that can be investigated for probiotic potential. We also used 16S rRNA gene repertoire analysis to investigate the cecal microbiota taken from 7-day-old chicks from the two conventional high performance farms but also from a free-range farm presenting lower performance indicators. We choose this young age since it has been described that intestinal microbiota start to shift from facultative to strict anaerobes approx. 1 week after birth, and that early exposure has important impact on adult microbiota composition [24]. Due to the greater abundance of *C. jejuni*-related sequences in the caecal content of low-performing chicks collected from the free-range farm, the commensal strains isolated from high-performing farms were finally tested for their ability to inhibit the growth of *C. jejuni* under conditions reflecting the life cycle of chickens.

## Results

### 16SrRNA gene-repertoire analysis of high performer and low performer farms

In total, 5,328,123 reads were generated and analyzed in this study. Median and mean sequence coverage were 169,697 and 177,604 reads per sample, ranging from 114,571 to 250,671 reads in the sample with the lowest and the highest coverages, respectively. 6,105 OTUs were identified, of which 400 were supported by more than 0.005% reads. The 16S rRNA gene-repertoire was analysed for ten caecal contents collected from high- (Farms 1 and 2) or low- (Farm 3) performing farms. A total of 5 phyla were identified based on 16S rRNA gene repertoire analysis, with substantial variation in the abundance among individual microbiomes. Sequences that were affiliated to the Firmicutes phylum represented at least 70% of the sequences in all analyzed caeca (S1 Fig). The only exceptions were for caeca #5 and #7 collected from Farm #1 (high performers) for which Bacteroidetes were present in high proportions (23.6% and 35.1%, respectively), and caecum #4 collected from Farm #3 (low performers) for which Proteobacteria (family: *Enterobacteriaceae*) were present in high proportions (31.3%) (Fig 2). Overall, 0.9% to 13.5% of sequences were affiliated to Proteobacteria in caeca collected from Farm #1, 3.9% to 29.9% from Farm #2 and 0.6% to 31.3% from Farm #3. Sequences affiliated to the phyla Actinobacteria and Tenericutes were almost exclusively detected in caeca collected from Farm #3, with different abundances among individuals (Actinobacteria: 0.2 to 4.3%, Tenericutes: 0.6 to 1.9%). Sequences affiliated to the phylum Bacteroidetes represented 0.1% to 35.1% of the 16S rRNA gene repertoire for caeca collected from Farm #1, and less than 1% for caeca collected from Farm #2 and Farm #3, respectively (S1 Fig). At the family level, *Lachnospiraceae*, *Ruminococcaceae*, *Lactobacillaceae* and *Enterobacteriaceae* were the most abundant in all caeca collected from the 3 farms (S1 Table). When

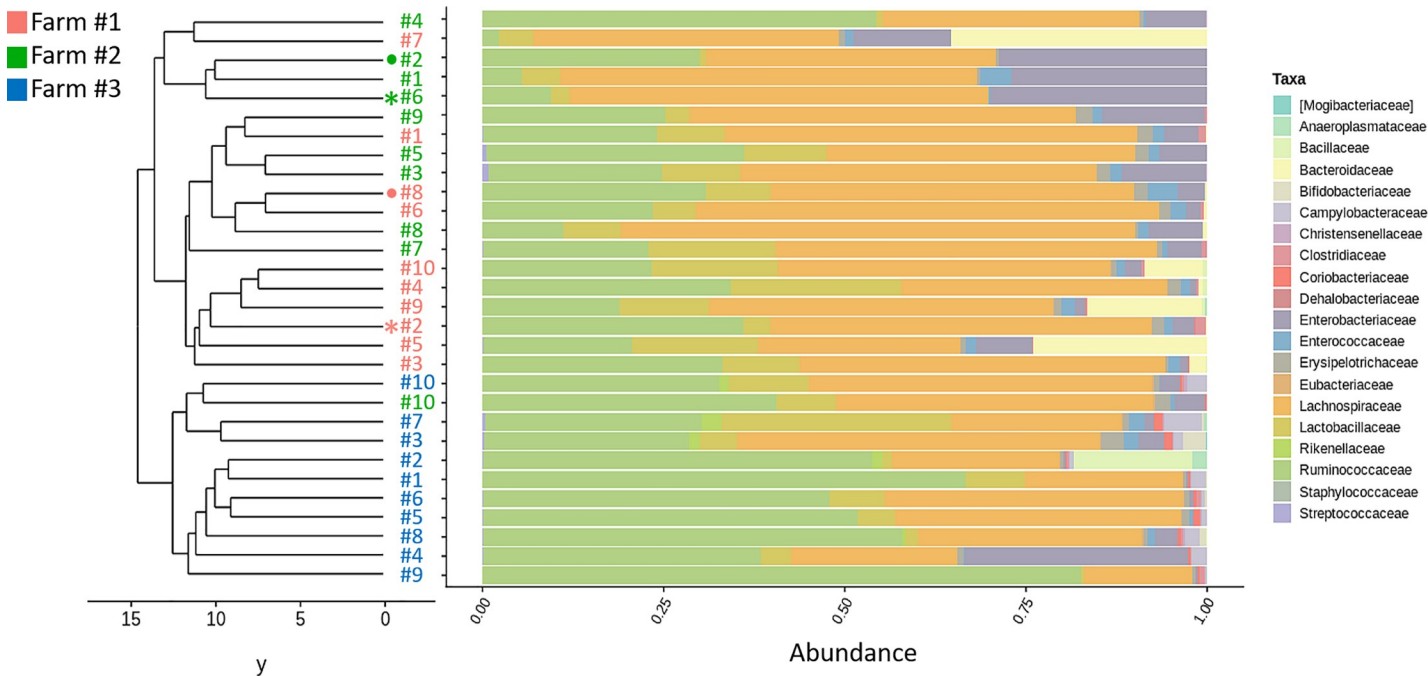

**Fig 1. Relative abundance and clustering of 30 samples colored by farm origin.** Samples were clustered using UPGMA with Camberra distance. Colored bars represent the relative abundance of bacterial family.

performing phylogenetic clustering at family level, samples collected from Farm #3 clustered together whereas there was no clear distinction among samples collected from Farms #1 and #2 (Fig 1). *Bacteroidaceae* were detected in high numbers only in 4 of 10 samples collected from Farm #1, representing in average 8.7 ± 12.7% of OTUs for this farm. Sequences affiliated to the *Campylobacteraceae* family were detected only in samples collected from Farm #3 (low performers).

At the genus level, sequences affiliated to species historically included in the *Ruminococcus* genus but that belong to the *Lachnospiraceae* family based on recent phylogenetic analysis were the most frequent, representing 32.6%, 30.5% and 21.0% of the sequences in caeca collected from Farms #1, 2 and 3, respectively (Fig 2, indicated as [*Ruminococcus*]). Sequences affiliated to the *Lactobacillus* genus were also frequent in samples collected from the three farms (15.4%, 10.2% and 11.5% of the sequences in caeca collected from Farms #1, 2 and 3, respectively), as well as sequences affiliated to the *Oscillospira*, *Ruminococcus* (family: *Ruminococcaceae*), *Blautia*, *Escherichia*, *Butyciricoccus*, and *Coprococcus* genera that were recovered in relatively high proportions (> 2%) in caeca collected from the three different farms (Fig 2). Sequences representing more than 1% of the relative abundances and that were differentially abundant in high *vs* low performer groups were also detected. They belonged to the genera *Faecalibacterium* and *Campylobacter* that were significantly ($P< 0.05$) enriched in the low performers group, and to the genera *Bacteroides*, *Proteus*, *Enterococcus*, *Escherichia* and *Klebsiella* that were significantly ($P< 0.05$) enriched in the high performers groups.

## Diversity and richness of the caecal microbiota

The diversity (Shannon index) was significantly higher ($P<0.05$) in the caeca collected from the low performer chicks than in caeca collected from the high performer chicks (Table 1). The total number of bacterial genera and predominant genera was not different among the 3

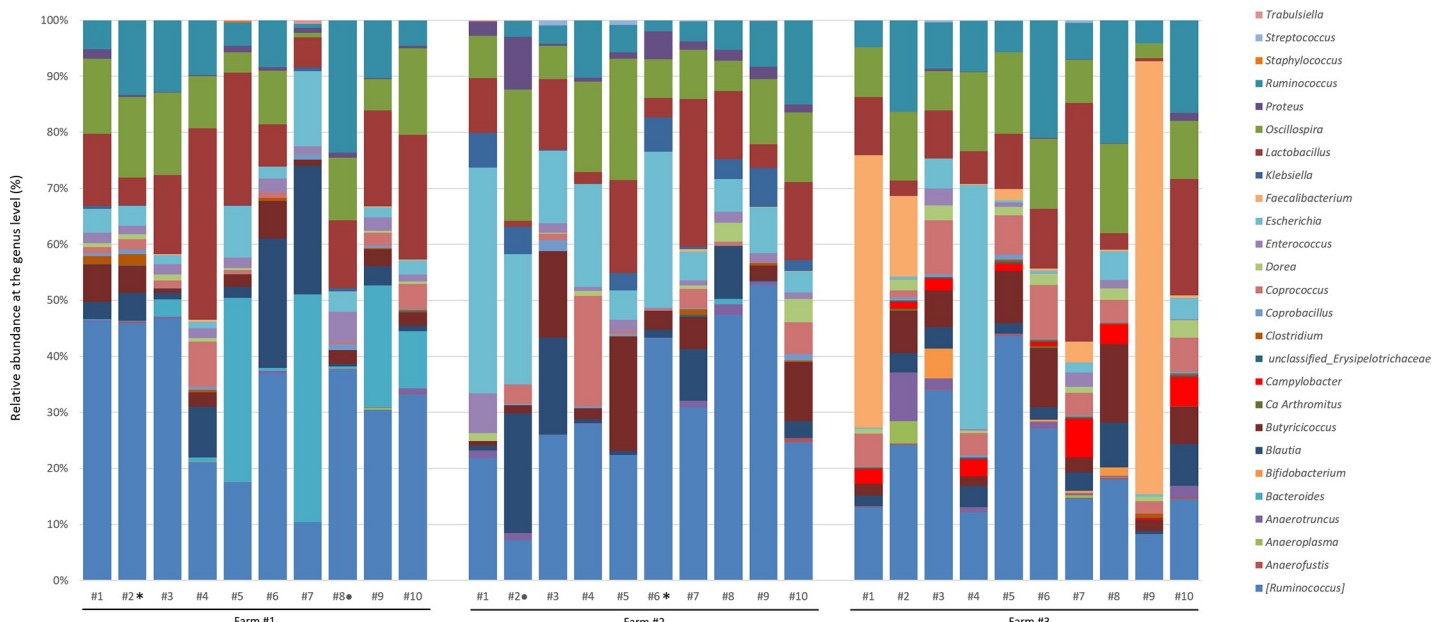

**Fig 2. Genus-level 16S rRNA gene-repertoire analysis of caeca from high and low performer farms.** The bacterial genera composition of each sample was obtained from the taxonomic annotation method RDP.

different farms. However the observed sample richness at the OTU level was significantly higher ($P<0.05$) in the caeca collected from the low performer chicks than in caeca collected from the high performer chicks (Fig 3).

On the first component of the PCoA with Bray Curtis distance, samples from Farm #1 and Farm #2 were closer to each other than they were to samples from Farm #3, suggesting that samples from Farm #1 and Farm #2 have more similar microbiota (Fig 4). Conversely, samples from Farm #3 were isolated, suggesting underlying differences in composition. A Permanova test was also performed under the null hypothesis that the mean and the dispersion of both groups are equivalent. This test discriminates groups based on one or more factors (explicative variables). In this work, performance between farms (high/low) was used as the only factor. A significant difference was found between high- and low- performance farms with a *P*-value of 0.001.

## Isolation, cultivation and identification of commensal bacteria

For each high performer farm (Farm #1 and Farm #2), two caeca were randomly chosen from the 10 received caeca. The caecal samples were processed with or without preselection treatments to cultivate commensal bacterial species. A total number of 347 isolates was obtained from Farm #1 and 234 from Farm #2, of which 218 (63.8%) and 140 (59.8%) could respectively be sub-cultivated after primary cultures isolation (Table 2).

**Table 1. Caecal bacterial diversity and richness of 7-days old chicks.**

|                                  | Farm #1    | Farm #2    | Farm #3    |
|----------------------------------|------------|------------|------------|
| Richness index                   | 214 ± 28   | 195 ± 34   | 273 ± 29   |
| Shannon index                    | 3.5 ± 0.4  | 3.5 ± 0.3  | 3.9 ± 0.6  |
| No. of genera                    | 21 ± 2     | 19 ± 2     | 22 ± 1     |
| No. of predominant genera (>1%)  | 8 ± 1      | 9 ± 2      | 9 ± 2      |

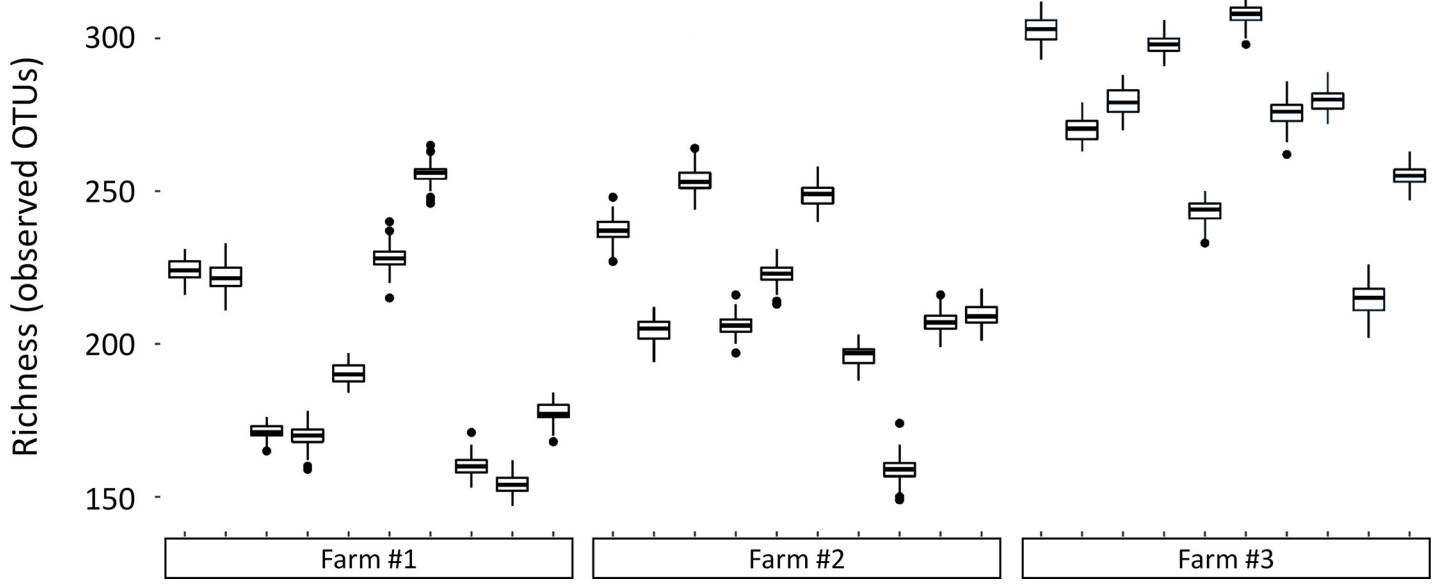

**Fig 3. Sample richness (observed OTUs).** One-hundred subsamplings of 33,654 sequences per sample were used to estimate the richness variability.

**Fig 4. Beta diversity analysis.** PCoA for ordination and Bray curtis distance.

**Table 2. Isolates obtained from each high performer farm using different culture conditions.**

| Culture media and selection treatments | Farm #1 | | Farm #2 | |
|---|---|---|---|---|
| | Primary culture | Sub-culture | Primary culture | Sub-culture |
| mGAM +/- antibiotics and rumen fluid | 198 | 140 | 113 | 67 |
| LB after heat treatment | 13 | 11 | 13 | 11 |
| LBS | 3 | 2 | 15 | 9 |
| mGAM + TCA +/- rumen fluid | 133 | 65 | 93 | 53 |
| Total | 347 | 218 | 234 | 140 |

Local spectra database was implemented with the spectrum of every analyzed isolate. All the spectra were then compared to each other and isolates were considered as belonging to the same species when spectra homologies were higher than the settled threshold of 80%. For Farm #1, 118 (54.1%) of the isolates could not be identified due to absence of reference spectra in the database, and the identification was not possible for 48 (22.0%) of the isolates due to poor quality spectra. Between 1% and 5% of the analyzed isolates were identified as *Clostridium paraputrificum*, *Enterococcus gallinarum*, *Enterococcus hirae*, *Escherichia coli* and *Lactobacillus crispatus*, whereas other identified isolates accounted for less than 1% of isolates analyzed for Farm #1 (S2 Table). For Farm #2, 66 (47.1%) of the isolates could not be identified due to absence of reference spectra in the database and poor quality spectra were obtained for 18 (12.9%) of the isolates. *E. coli* represented 16.8% of the analyzed isolates whereas *Clostridium ramosum*, *Enterococcus avium/raffinosus*, *Klebsiella pneumoniae*, *L. crispatus* and *Proteus mirabilis* accounted for 1 to 5% of isolates analyzed for Farm #2. Other identified isolates accounted for less than 1% of isolates analyzed for Farm #2 (S2 Table).

One, randomly chosen isolate from a group of homologues with or without species assignment (due to the absence of reference spectrum or low quality spectra) was retained for 16S rRNA gene-based identification. Partial 16S rRNA-encoding gene sequences were obtained for 241 isolates and blasted against the NCBI nucleotide database. When taking into account all the isolates that fall in the same MALDI-TOF identification groups, the vast majority of the isolates (304 of 352, i.e. 86.4%) belonged to the Firmicutes, whereas Proteobacteria represented 13.4% (47 of 352) and Actinobacteria only 0.3% (1 isolate) of the total isolates (S3 Table). Partial 16S rRNA gene identities with sequences of the closest described species were ≥ 99% (labelled as '++' in Fig 5 and S3 Table) for all unique sequences affiliated to the Actinobacteria and to the Proteobacteria. Among unique sequences corresponding to isolates affiliated to the phylum Firmicutes, the vast majority of those affiliated to the *Paenibacillaceae*, *Bacillaceae*, *Enterococcaceae* and *Lactobacillaceae* families also had ≥ 99% identities with sequences of the closest described species (Fig 5 and S3 Table). This was also the case for all except 2 sequences (corresponding to a total of 5 isolates in the MALDI-TOF identification groups) affiliated to the *Erysipelotrichaceae* family. Four of the 22 unique partial 16S rRNA gene sequences affiliated to the *Ruminococcaceae* family, corresponding to 22 isolates based on MALDI-TOF grouping, presented only ≥95 to <97% identities with 16S rRNA gene sequences of the closest described species. Fifteen of the 26 unique partial 16S rRNA gene sequences affiliated to the *Lachnospiraceae* felt in the ≥95 to <97% identity category, 5 of 26 in the ≥93 to <95% identity category and 1 in the < 93% identity category. This last one is likely to represent a new species or even a new genera due to low homology with already described species.

As expected, *Bacillaceae* and *Paenibacillaceae* isolates were cultivated from heat-treated caecal samples, with the exception of several *Bacillaceae* isolates mainly corresponding to *B. licheniformis* that were recovered after ethanol treatment (S3 Table). Several *Enterococcaceae* isolates were recovered from heat-treated samples but most were from LB and mGAM agar

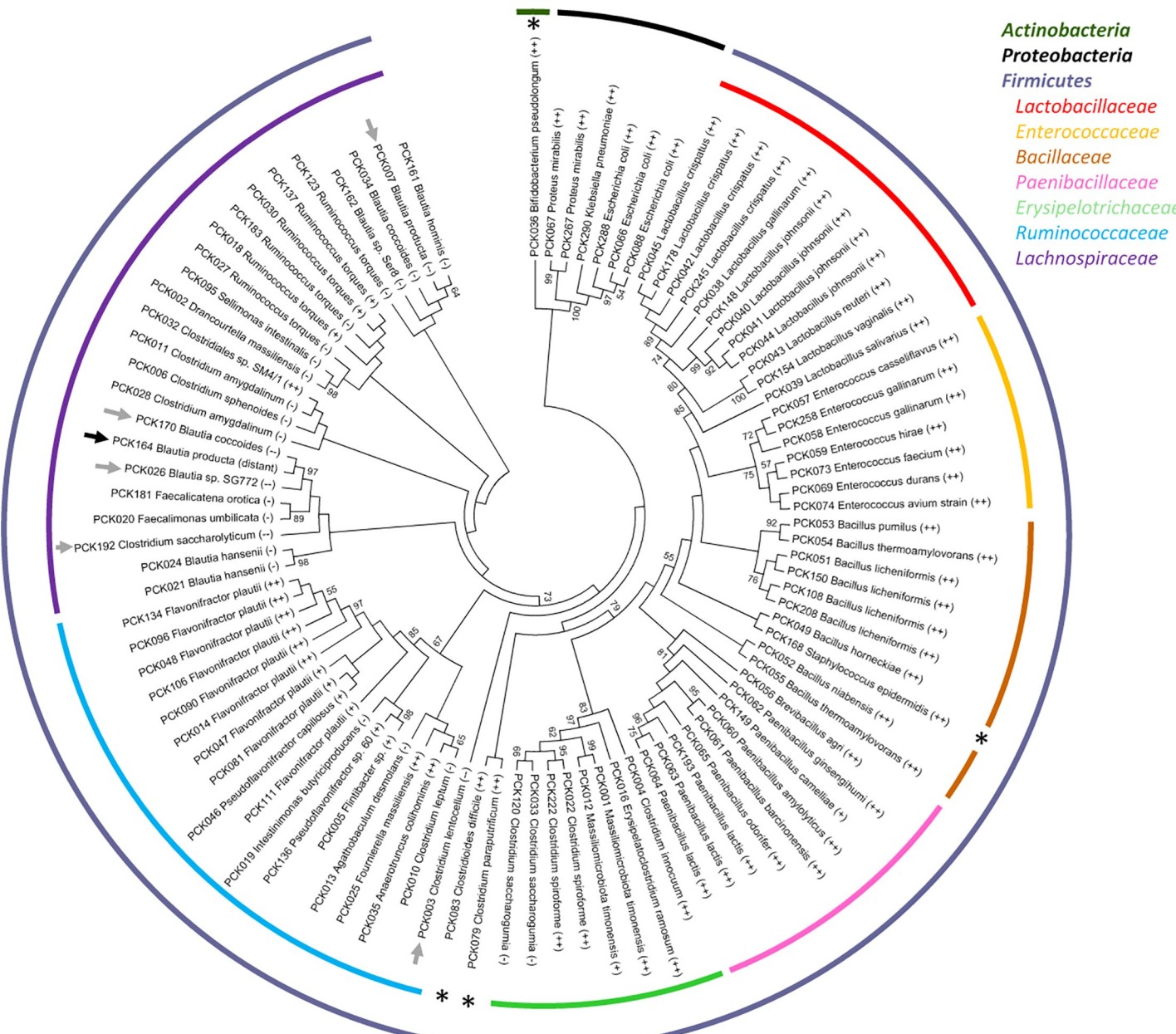

**Fig 5. Phylogenetic tree representing commensal species isolated from high performer chicks.** The phylogenetic tree was inferred from Muscle alignment of partial 16S rRNA-encoding gene sequences using the Maximum Likelyhood method based on the Kimura 2-parameters model with 1,000 bootstrap replicates. Identifications were classified according to the identity percentage as: 1) '++' for ≥ 99–100% identity, 2) '+' for 97–98% identity, 3) '-' for 95–96% identity, 4) '—' for 93–94% identity and 5) 'distant' for < 93% identity. Stars indicate families for which only one unique partial 16S rRNA gene sequence was recovered. Arrows indicate isolates likely to correspond to new species (grey arrows) or new genera (black arrows) based on limited 16S rRNA gene identity with the closest described species.

plates containing various antibiotics. Several *Erysipelotrichaceae* were recovered from ethanol-treated samples but most were recovered from mGAM and mGAM + rumen fluid with or without sodium taurocholate and in the presence of various antibiotics. This was also the case for isolates affiliated to *Lachnospiraceae*, whereas more than half of the 104 *Ruminococcaceae* isolates, corresponding to 13 unique 16S rRNA gene sequences, were recovered only from eth-anol-treated samples (S3 Table). Seven isolates corresponding to *Clostridium paraputrificum*

(family: *Clostridiaceae*) and 2 corresponding to *Clostridioides difficile* (family: *Peptostreptococcaceae*) were recovered from ethanol-treated samples seeded on mGAM plates with or without rumen fluid and sodium taurocholate. All isolates affiliated to *Lactobacillaceae*, *Enterobacteriaceae*, *Morganellaceae*, *Staphylococcaceae* and *Bifidobacteriaceae* were recovered from non-treated caecal samples seeded on various plates with or without antibiotics.

Overall, bacterial strains corresponding to the OTU detected by analysis of the 16S rRNA repertoire were recovered for most of the predominantly abundant families, with the exception of *Bacteroidaceae* (Fig 6). Cultivated strains represented 8.5% of total OTUs detected in the 16S rRNA-encoding gene repertoires of the two different farms, and these cultivated OTUs accounted for 49.2% relative abundance over total OTUs.

## Full genome sequencing of *Campylobacter jejuni* field isolate

Approx. 1.8 kb were assembled after sequencing, with 97.8% of the reads that could then be mapped on the assembled genome. The tool BUSCO [25] was used to compare gene content of the new genome with reference genome of *C. jejuni* GCF_000009085.1, resulting in 93.3% of the genes being recovered in single copy and 6.4% being duplicated. The tool GTDB-Tk [26] was used to calculate the average nucleotide identity (ANI) value between the genome sequence and *C. jejuni* reference genome, resulting in a 97.59% ANI value which confirms that the new isolate belongs to the species *C. jejuni*.

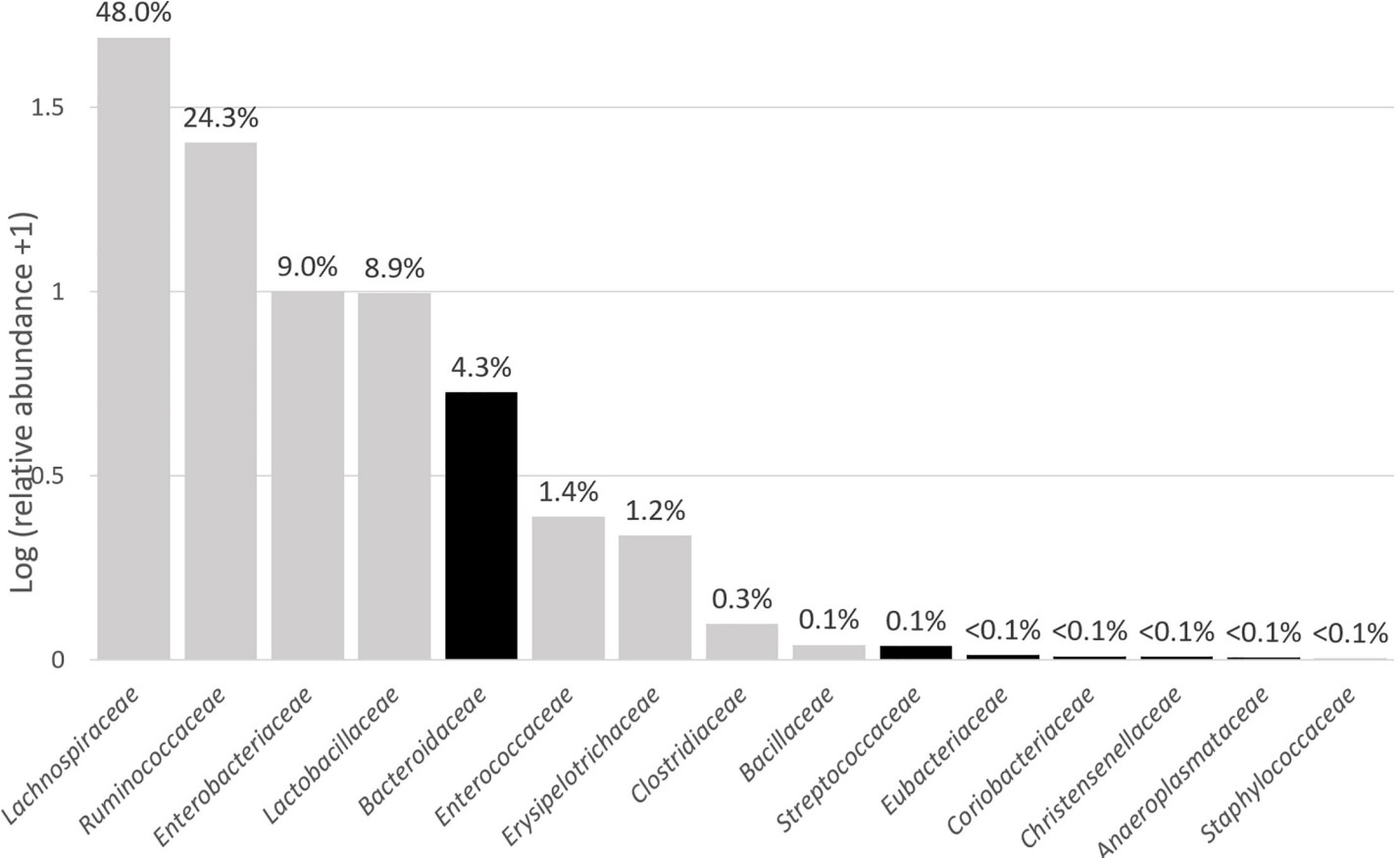

**Fig 6. Mean abundancies of families detected in cecal contents of high-performance chicks using 16S repertoire analysis.** Data are represented as Log (relative abundance +1) for better visualization, with corresponding abundancies being indicated on top of each bar. Families for which bacterial isolates affiliated to the family were recovered in culture are indicated in grey, those for which no bacterial isolates affiliated to the family were recovered in culture are indicated in black.

### Antibiosis activity against *Campylobacter jejuni*

Antagonistic activity against several *C. jejuni* strains was tested for a selection of commensal strains isolated from high performer chicken. We first tested a selection of 26 aerobic/facultative anaerobic strains cultivated at 37˚C in aerobic conditions against the reference strain *C. jejuni* ATCC 700819. When antagonistic activity was detected, the strains were further tested after cultivation at 42˚C in aerobic conditions, as well as cultivation at 37˚C or 42˚C in anaerobic conditions. In addition the strains presenting activity against *C. jejuni* ATCC 700819 were tested against the strain isolated from Farm #3. A total of 7 strains presenting significant activity and 2 presenting moderate activity were identified, including 4 *Enterococcus* species, 3 *Lactobacillus* species, 1 *Bacillus* species and 1 *E. coli* strain (Table 3 & S2 Fig). Enterococci were the most active, with activity against the reference strain being detected in all tested conditions. However the *C. jejuni* strain isolated from Farm #3 was slightly more resistant than the ATCC strain, which was observed when Enterococci strains were cultivated at 42˚C instead of 37˚C, especially in aerobic conditions. The activity of the other commensal strains was more limited, being mostly observed when bacteria were cultivated at 37˚C in aerobic conditions, with the exception of two *Lactobacillus* strains that were also active when cultivated at 42˚C in aerobic and/or anaerobic conditions. The 32 commensal anaerobic strains tested after growth at 37˚C (all strains) or 42˚C (18 of the 32 strains) in anaerobic conditions did not display any antagonistic activity against *C. jejuni* ATCC 700819 (S5 Table). The only exception was for the strain PCK036 identified as *Bifidobacterium pseudolongum* but this was likely due to medium acidification as demonstrated when incorporating phenol red in the plate.

## Discussion

Gut microbiota composition of livestock animals has been linked to resistance to infections and improved feed conversion, with competitive exclusion of pathogenic bacterial species by the normal resident microbiota being reported in the early 1970s [27]. With the advent of high-throughput sequencing, it is now possible to explore the composition of complex microbial populations in a relatively simple and cost-effective way, with the objective to better understand and take profit of the beneficial effects conferred by specific commensal microbiota. With this objective in mind, we chose to compare caecal microbiota composition of 7-days old chicks raised in high- *vs* low-performing farms, and then cultivated a variety of

**Table 3. Antibiosis activities against *C. jejuni* ATCC 700819 and Farm #3 isolate detected in commensal strains.** Inhibition diameters were measured and scored with the following parameters: (+) for inhibition diameters ≥ 2 mm, (+/-) for 0.1 to 1.9 mm, (-) when no inhibition was observed. nd: no done due to the absence of growth in tested conditions.

| Aerobic/facultative anaerobic strains tested | Antibiosis activity against *C. jejuni* | | | | | | | |
| --- | --- | --- | --- | --- | --- | --- | --- | --- |
| | ATCC 700819 | | | | Farm #3 isolate | | | |
| | Aerobic | | Anaerobic | | Aerobic | | Anaerobic | |
| | 37˚C | 42˚C | 37˚C | 42˚C | 37˚C | 42˚C | 37˚C | 42˚C |
| PCK178_*Lactobacillus crispatus* (++) | (+) | nd | (-) | (+) | (-) | nd | (-) | (+) |
| PCK039_*Lactobacillus salivarius* (++) | (+) | (+) | (-) | (-) | (+) | (+) | (-) | (-) |
| PCK040_*Lactobacillus johnsonii* (++) | (+/-) | nd | (-) | (-) | (-) | nd | (-) | (-) |
| PCK049_*Bacillus horneckiae* (++) | (+/-) | (-) | nd | nd | (-) | (-) | nd | nd |
| PCK066_*Escherichia coli* (++) | (+) | (-) | (-) | (-) | (+) | (-) | (-) | (-) |
| PCK057_*Enterococcus casseliflavus* (++) | (+) | (+) | (+) | (+) | (+) | (-) | (+) | (+) |
| PCK058_*Enterococcus gallinarum* (++) | (+) | (+) | (+) | (+) | (+) | (+/-) | (+) | (+) |
| PCK069_*Enterococcus durans* (++) | (+) | - | (+) | (+) | (+/-) | (-) | (+/-) | (-) |
| PCK074_*Enterococcus avium* (++) | (+) | (+) | (+) | (+) | (+) | (+) | (+) | (+) |

commensal species from the samples collected from high-performing farms. 16S rRNA gene repertoire sequencing results demonstrated significant variations in each of the three groups, which was also reported by others in studies including higher numbers of individuals and hypothesized to be linked to lack of colonization of the young chicks by maternally derived bacteria and high hygiene levels maintained in hatcheries [28]. However, since inter-group variations were more significant than inter-individual variations within the groups, we were able to compare the three groups. We also compared our results with those obtained by others even if it should be kept in mind that inter-studies comparisons are difficult to conduct due to differences in biological (chickens lines, ages, nutrition, treatments. . .) or technical (DNA extraction, sequencing technologies and associated bioinformatics. . .) parameters.

Overall Shannon's index of diversity was low, which was not unexpected for such young animals for which a gradual shift from facultative anaerobes (mainly *Enterobacteriaceae*) to strict anaerobes (mainly *Clostridiales*) as well as an increase in caecal microbiota diversity is then observed with age [24]. The poor-performing farm was an organic farm in which chicks had access to an outdoor range, which can likely explain the fact that richness and diversity were higher for animals raised in contact with increased bacterial diversity compared to animals raised in conventional farms. At the family level, 16S rRNA gene repertoire results observed in our study are in line with results usually reported by others, with *Lachnospiraceae*, *Ruminococcaceae*, *Lactobacillaceae* and *Enterobacteriaceae* being often dominant in young chicks' caecal microbiota [24, 29, 30]. At the genus level, the microbial composition of high performers' caeca was also close to compositions reported for similar age chicks [29]. The most notable differences between high and low performers were higher abundances of OTUs related to *Faecalibacterium* and *Campylobacter* in low- compared to high-performers. *Campylobacter* colonization was confirmed by isolating and cultivating *C. jejuni* from these animals and then sequencing the genome of the isolate. Peak in *C. jejuni* prevalence has been reported to occur between 10 and 13 weeks of age for free-range chickens [31] but colonization can potentially appear around 2 weeks of age, when a shift in gut microbiota composition could create a window opportunity [19]. *C. jejuni* has been reported to trigger prolonged inflammatory response and damage to mucosa in susceptible birds, resulting in significant negative impact on birds' welfare in commercial production [18]. Since it has been reported that the potentially beneficial species *Faecalibacterium prausnitzii* has anti-inflammatory effects [32, 33], its increased abundance might improve the ability of *Campylobacter*-colonized chicks to control inflammation, while consuming more energy as a cost [34]. Conversely, it has also been reported that Short Chain Fatty Acids (SCFAs), and especially butyrate can trigger the expression of *C. jejuni* genes that are important for host colonization, whereas lactate which is present in higher concentrations in the upper part of the digestive tract has an inhibitory effect on this expression [22, 23]. In our study, *F. prausnitzii*, which is a well-known butyrate producer [35, 36], was only detected in the low-performers chicks colonized by *C. jejuni*, where as *Enterococcaceae* that are well-known lactate producers were more abundant in the high-performers chicks that were not colonized by *C. jejuni*. Interestingly, the family *Bifidobacteriaceae* to which belong species that were described to enhance butyrate formation by cross-feeding acetate to *F. prausnitzii* [37] was also only detected in the low-performers chicks. Whether or not these correlations are relevant and could play a role in *C. jejuni* colonization through cross-feeding mechanisms will be investigated in future studies.

The second part of our study consisted in an effort to isolate commensal species from good performance farms that could then be mined for specific phenotypes making them good probiotic candidates. Since we were mainly interested at isolating commensal bacteria that could potentially be used as probiotics, we did not use specific methods for the recovery of fastidious taxa. These cultivation experiments were conducted with caecal contents collected from the

same animals from which caeca were collected for the 16S rRNA gene repertoire study, so both studies (sequencing and cultivation) were run in parallel. We succeeded at isolating a variety of strains representing 8.5% of the 6,105 OTUs detected using 16S rRNA repertoire analysis, most of these strains being affiliated to the most abundant families (relative abundance >1%) in these animals, with the notable exception of members of the *Bacteroidaceae* family. Failure to recover isolates affiliated to the *Bacteroidaceae* is likely due to the fact that corresponding OTUs (all affiliated to the *Bacteriodes* genus) were detected in significant amounts only in cecal samples collected from farm #1, and that they were not equally distributed, with 4 samples presenting relative abundances >10% and 5 with relative abundances <1%, the last one being at 3.1%. Farm #1 sample #2 that was randomly chosen for cultivation without ethanol pre-treatment contained only 0.1% of OTUs affiliated to the *Bacteriodes* genus, thus explaining why we were not able to recover members of this family in culture. *Bacteroides* species are strictly anaerobic and do not form spores, so they are unlikely to be used as probiotics and we have therefore chosen not to put more effort on this point. Cultured isolates include well described species that belong to the *Bacilli* class (families: *Bacillaceae*, *Paenibacillaceae*, *Lactobacillaceae* and *Enterococcaceae*). Not surprisingly, higher proportions of isolates likely corresponding to new species or even new genera were present in those affiliated to the classes *Clostridia* (families: *Lachnospiraceae* and *Ruminococcaceae*) and *Erysipelotrichia* (family: *Erysipelotrichaceae*). This novelty was mainly found in isolates affiliated to the *Lachnospiraceae* family (strains PCK003, 007, 026, 170, 192 labelled '—', and strain PCK164 labelled 'distant' in Fig 5), suggesting that only limited numbers of strains corresponding to this family were cultivated so far. This was also the case in a recent study published by others [38]. Interestingly, in this extensive work the authors used a very similar culturomics approach than the one that we used in our study, with the exception that only Wilkins-Chalgren anaerobe agar (WCHA) supplemented with rumen fluid and several additional compounds was used for cultivation of commensal caecal bacteria from chicken of 4 to 40 weeks of age. Heat as well as ethanol treatments were also used to inactivate vegetative cells, and antibiotics were incorporated in the medium to select specific bacterial species. Sodium taurocholate was omitted. This bile acid has been described to be a potent germinant for all spore-formers and many gut commensal species have the capacity to form spores that resist ethanol exposure [39]. In our hands, adding it at 0.1% final concentration in mGAM + 30% rumen fluid resulted in 1 to 2 log higher CFU counts on plates seeded with ethanol-treated caecal samples. Other culturomics studies that did not include spores selection treatments were less successful at isolating a diversity of commensal species from similar samples [40]. Of note, we found heat pretreatment to be useful for the selection and recovery of *Enterococcaceae* in addition to sporulated *Bacillaceae* and *Paenibacillaceae*, which could be explained by relatively high heat resistance reported for selected species/strains in this family. In their recently published culturomics study, Crhanova *et al* used 174 different culture conditions and succeeded at culturing 42% of gut microbiota members [41]. This already elevated recovery rate could potentially be improved by incorporating rumen fluid in the culture media as recently reported by others [42]. However this material is difficult to source and to prepare, and its inherently variable composition can result in variable culture efficacy.

The production of bacteriocins active against *C. jejuni* has been demonstrated for several lactic acid bacteria (LAB) including *Enterococcus* and *Lactobacillus* species, as well as for *Paenibacillus* and *Bacillus* species, and administration of purified bacteriocins can reduce colonization by *C. jejuni* [43–45]. Non-bacteriocin compounds produced by *Enterococcus faecalis* and *Lactobcillus reuteri* and active against *C. jejuni* have also been reported [16, 46]. In addition to LAB and *Bacillus* species, strictly anaerobic commensal species including *Blautia* (family: *Lachnospiraceae*) and *Ruminococcus* (family: *Ruminococcaceae*) species have also recently been

demonstrated to produce bacteriocins active against a variety of pathogenic species [47, 48]. In an attempt to select candidate probiotic species from high-performer chickens, we thus tested LAB, *Bacillus* species and a selection of *Erysipelotrichaceae*, *Ruminococcaceae* and *Lachnospiraceae* for the production of compounds active against *C. jejuni*, including the *C. jejuni* strain that was isolated from low-performer animals. These tests were performed with bacteria grown under conditions that are expected to reflect the developmental stages of the host and its intestinal microbiota. The temperature at which eggs are incubated in hatcheries is also close to 37˚C, and 42˚C is the physiological temperature of the birds. At birth, the gut of young animals is first colonized by facultative anaerobes (mainly *E. coli*) which then create a new environment that promotes the colonization by strict anaerobes. Significant differences were observed depending on conditions used to cultivate candidate anti-*C. jejuni* compounds producers: *L. crispatus* PCK178 displayed significant activity only when cultivated at 42˚C in anaerobic conditions, whereas *L. salivarius* PCK039 and *E. coli* PCK066 displayed activity only when cultivated at 37˚C in aerobic conditions (Table 3 and S2 Fig). Differences were less pronounced for other isolates presenting an activity. Additional experiments are needed to better understand if observed antagonistic activities are due to bacteriocins or to other kinds of compounds [16]. The fact that none of the 32 strictly anaerobic isolates tested displayed detectable antibiosis activity was not completely unexpected. In their recent study, Kim *et al* screened 421 human commensal isolates including *Blautia* and *Ruminococcus* species and could detect only 2 presenting antibiosis activity against *E. faecium* [47]. Similarly, screening a collection of 17 *F. prausnitzii* strains did not reveal any antibacterial effect on several anaerobic and aerobic bacterial species [49].

In conclusion, our study confirms the differences in the composition of caecal microbiota in young chickens depending on the farm. The relatively limited number of samples tested makes it difficult to draw any conclusions about the association of certain bacterial groups with the presence of *Campylobacter* sp. in low-performing chickens. However, these results pave the way for more specific studies in which metabolomic analyses will also have to be included to explore the association between the presence of certain bacterial groups and the presence of certain metabolites that inhibit or promote colonization by *Campylobacter* sp. The culturomics methods used have proven to be very efficient, allowing the culture of a wide variety of commensal bacterial species in chicken. It was important to take into account temperature and oxygenation conditions similar to those found at different stages of animal development during the exploration of *C. jejuni*- antagonistic activities by these species. These parameters should make it possible to better select potential probiotics according to their intended use and in particular their administration *in ovo* or *in vivo* at different ages.

## Materials and methods

### Farms selected for the trial

This study was carried out in strict accordance with the recommendations of the Agence nationale de sécurité sanitaire de l'alimentation, de l'environnement et du travail (ANSES). The protocol was approved by the local Committee on the Ethics of Animal Experiments of Boehringer Ingelheim Santé Animale Research Center (Protocol Number: 16.0140.P). All surgery was performed under sodium pentobarbital anesthesia, and all efforts were made to minimize suffering.

Caecal samples were collected from 7-days old chicks coming from three different poultry farms integrated by the same company. Two farms were classified as high performers and one as low performer (Table 4) based on historic, significantly different and stable performances in terms of average birds body weight, mortality and average daily weight gain. The source of chicks was the same for farms 2 and 3, whereas it was different for farm 1. Stocking density was identical for farms 1 and 2 whereas chicks were raised in free-range for farm 3. Chicks

**Table 4. Characteristics of the poultry farming.**

| Poultry farming | Farm #1 | Farm #2 | Farm #3 |
|---|---|---|---|
| Classified as | High performer | High performer | Low performer |
| Chickens' strain | ROSS 308 | ROSS 308 | ROSS 308 |
| Average birds body weight (kg) | 1.925 | 1.980 | 1.954 |
| Mortality (%) | 3.86 | 4.78 | 10.00 |
| Average daily weight gain (g/day) | 56.54 | 56.92 | 52.81 |
| Feed Conversion Ration (FCR) | 1.668 | 1.781 | 1.832 |
| Water treatment | Hydrogen peroxide | Chlorine dioxide | none |

from the three farms were fed with the same diet composed of wheat, corn, soya meal, sunflower meal, rapeseed meal, soya oil, sodium sulfate, phosphate as well as additives including vitamins, oligo-elements, amino acids and digestibility enhancers. Animals did not receive any antibiotic treatment before the sampling.

## Chemicals and culture media for isolation and cultivation of commensal bacteria

All the antibiotics used in this study as well as the sodium taurocholate were obtained from Sigma-Aldrich except Neomycin that was obtained from Oxoid. Final antibiotics concentrations were taken from previously published studies [50, 51] and are reported in Table 5. Cultivation media used in these experiments were (1) modified Gifu Anaerobic Medium (mGAM, Hyserve), (2) Luria-Bertani (LB) Miller medium (Difco) or (3) *Lactobacillus* medium (LBS, BD). Rumen fluid collected from healthy cattle receiving winter pasture feeding was also used as source of undefined growth factors.

## Caecal sample collection

For each farm, ten 7-days old chicks were euthanized on-site with a lethal dose of Do lethal® administered by intra-occipital venous sinus injection. Both ceca were ligatured and collected from each chick, then placed into a GENbag (bioMérieux) with a generator sachet and delivered to the lab within 5 hours. Upon receipt, ceca were either (i) directly processed for the cultivation of non-sporulated, oxygen-sensitive bacteria and lactobacilli, or (ii) directly stored at -80˚C for subsequent cultivation of sporulated bacteria or before DNA extraction and 16S rRNA gene repertoire sequencing.

## DNA extraction and 16S rRNA gene repertoire sequencing of caecal contents

DNA was extracted using the MoBio PowerMag Powersoil kit with a modified protocol. For each sample, approx. 50 mg of cecal content were transferred in the bead beating tubes

**Table 5. Antibiotics and concentrations used to selectively grow a variety of gut commensal species.**

| Antibiotics | Final concentration in cultivation media (µg/ml) |
|---|---|
| Erythromycin (ERY) | 32 |
| Ciprofloxacin (CIP) | 16 |
| Gentamycin (GEN) | 32 |
| Vancomycin (VAN) | 32 |
| Sulfamethoxazole (SX) | 64 |
| Neomycin (NEO) | 75 |

provided with the kit. Bead beating was performed using the Retsch 400M at 30 cycles/sec (Retsch GmbH, Haan, Germany). Total extracted genomic DNA was quantified fluorometrically with the QuantiFluor ONE dsDNA System kit (Promega, WI, USA). Purity was determined by using the DropSense96 UV/VIS droplet reader (Trinean, Gentbrugge, BE). For each sample, DNA was diluted to 5 ng/μl which was then used in the PCR amplification step. Gene-specific primers for the 16S rRNA encoding gene (forward primer: 5'- CCT ACG GGN GGC WGC AG-3', reverse primer: 5'-GAC TAC HVG GGT ATC TAA TCC-3') were used to amplify the V3-V4 region. Primers were based on the Illumina's dual indexing sequencing principles of Illumina. Amplified PCR products were purified and normalized using the SequalPrep Normalization Plate Kit (Life technologies, CA, USA). Library size was controlled using the Agilent TapeStation HS1000 Screen Tape (Agilent technologies, USA) and final concentrations were measured using a SYBR green qPCR assay with primers specific to the Illumina adapters (Kapa Q-PCR Universal Library Quantification, Kapa Biosystems, USA). Libraries were denatured into single stranded molecules with freshly made 0.2N NaOH and diluted at 12 pM before being mixed with 30% of Illumina PhiX control libraries. Mixed Phix/16S libraries were sequenced in multiplex on the MiSeq machine with the MiSeq v3 chemistry to perform paired-end 300 bp sequencing. During sequencing the MiSeq was running Real Time Analysis software (RTA)version 1.18.54 and 2.5.05 MiSeq Control software. Sequence demultiplexing was performed automatically by MiSeq Reporter software version 2.5.

### Bioinformatics data analysis

The paired reads were assembled with FLASH [52], allowing for 25% of mismatches in the overlap region. Quality trimming and filtering (quality and length based) were performed with QIIME (v1.8) and prinseq (v0.20.3). The following parameters were used for this purpose: 75% of consecutive high-quality base calls; a maximum of 3 consecutive low-quality base calls; no ambiguous bases (N); a minimum quality score of 19. Chimeric sequences were predicted de novo with uchime which is integrated in the usearch 6.1 [53, 54]. The open-reference OTU picking approach was used for QIIME analysis with a sequence identity threshold of 97%. The following parameters were used with the Qiime pick_open_reference_otus.py pipeline: Green-Genes reference database (v13.8), usearch61 clustering method, 97% identity, taxonomy annotation method: RDP. Once the OTU table was generated a second level of sequence filtering based on OTU proportion (OTU proportion _ 0:005%) was performed as recommended by Bokulich *et al.* [55].

### Isolation and cultivation of commensal bacteria from high performers (S3 Fig)

Samples were handled in an anaerobic chamber (Bactron 600 –Shellab) filled with an atmosphere of 90% $N_2$ + 5% $H_2$ + 5% $CO_2$. Anaerobic atmosphere was verified with resazurin color indicators (BR0055 –Oxoid). Culture media, PBS and all other materials used for cultivation were placed in the chamber at least 48 h before use to reduce to anaerobic conditions. For the isolation of non-sporulated, oxygen-sensitive and facultative aerobic bacteria, one randomly chosen chick's caecum from each high performer farm (sample #2 for farm #1, sample #6 for farm #2) was cut open longitudinally and approx. 20 μl of cecal content was diluted in 3 ml of PBS. Ten-fold serial dilutions to $10^{-7}$ were performed. For the isolation of non-sporulated commensal bacteria, 100 μl of the last three dilutions were spread onto mGAM supplemented or not with antibiotics and then incubated at 37˚C under anaerobic atmosphere for a minimum of 5 days. Samples were processed the same way for the isolation of lactobacilli, and then

the dilutions were spread onto LBS and incubated at 37˚C under aerobic or anaerobic atmosphere for at least 2 days. After 2 to 5 days, colonies were subcultivated in mGAM or LBS for subsequent identification and storage. For the isolation of sporulated aerobic species, the remaining caecal suspensions were taken out of the anaerobic chamber and were then incubated at 80˚C for 20 min. Ten-fold serial dilutions were seeded on LB plates that were then incubated at 37˚C under aerobic atmosphere for 24 to 48 h. After 2 or 5 days, colonies were subcultivated in LB for subsequent identification and storage. For the isolation of sporulated, anaerobic and facultative anaerobic species, 20 µl of cecal contents from a high-performer chicken caecum stored at -80˚C (sample #8 for farm #1, sample #2 form farm #2) were diluted in 3 ml of PBS and then diluted with absolute ethanol to obtain final concentrations of 50% or 35% ethanol. As described elsewhere [39, 56], the tubes were incubated for 1 h (50% ethanol) or 4 h (35% ethanol) at room temperature under agitation at 33 rpm using a roller mixer. Two washing steps were performed with PBS to remove ethanol and the resulting pellet was re-suspended in 1 ml PBS and transferred in the anaerobic chamber. One-hundred µl of the $10^{-5}$, $10^{-6}$ and $10^{-7}$ dilutions were spread on the three different media: mGAM without antibiotics, mGAM + 30% rumen fluid, or mGAM + 30% rumen fluid + 0.1% sodium taurocholate. Plates were incubated at 37˚C under anaerobic atmosphere for a minimum of 5 days. After 2 or 5 days, colonies were subcultivated in mGAM + 30% rumen fluid for subsequent identification and storage. A *Campylobacter* strain was isolated from one low-performer chicken caecum stored at -80˚C. The caecum was thawed at 37˚C for 3 min, then diluted in PBS and seeded on the selective medium RAPID'Campylobacter Agar (BIORAD). Plates were incubated at 42˚C for 48 h in microaerophilic atmosphere (microAnaer bag).

## MALDI-TOF analysis for bacterial identification

VITEK MS (bioMérieux, France) was used for microbial identification and rapid de-replication of recurrent bacterial isolates. Colonies isolated from the various conditions were prepared using the direct transfer method from bioMérieux. Briefly, isolates were tested in duplicate by depositing one bacterial colony on the VITEK target slide, followed by either the addition of matrix solution (VITEK MS-CHCA) or formic acid (VITEK MS-FA). Evaporation and solvent crystallization of the matrix was performed at room temp. The loaded slide was then inserted into the VITEK MS system calibrated with the *Escherichia coli* ATCC 8739 standard. Microbial identification was achieved by comparing spectra with the VITEK MS database as well as with our own, user-built database. Micro-organisms were reported as a percentage of identity matching with the reference spectra. Isolates presenting a percentage of identity with reference species higher than 80% were considered as members of this species. One isolate corresponding to each isolate/group of isolates identified at the species level using MALDI-TOF analysis was randomly chosen for 16S-rRNA encoding gene sequencing. Isolates that did not present significant percentage of identity with reference spectra were grouped together (>80% spectra similarity with each other) and one isolate was then randomly chosen for phylogenetic identification using 16S-rRNA encoding gene sequencing.

## 16S rRNA gene sequencing of selected isolates

Single isolated colonies were cultivated in appropriate broth, and then the DNA was extracted from the pellets using the MoBio PowerMag Powersoil kit according to manufacturer's instructions. DNA was quantified using the NanoDrop 2000 (Thermo Scientific). Near full-length 16S rRNA gene, corresponding to regions V1-V9 was then amplified using primers 27F (5′-AGAGTTTGATCCTGGCTCAG-3′) and GM4R (5′-TACCTTGTTACGACTT-3′), and then sequenced using internal primers and the Sanger technology (GATC Biotech GmbH).

Sequences were BLASTED against the NCBI database with the exclusion of uncultured samples sequences. With the goal to easily represent robustness of phylogenetic assignation, identifications were classified according to the identity percentage as: 1) '++' for ≥ 99–100% identity, 2) '+' for 97–98% identity, 3) '-' for 95–96% identity, 4) '—' for 93–94% identity and 5) 'distant' for < 93% identity [57]. For phylogenetic analysis, 16S rRNA-encoding genes sequences were aligned using Muscle [58] integrated in MEGA7 [59] with default parameters. Only unique sequences that were different from each other were used for this analysis. The phylogenetic tree was inferred using the Maximum Likelihood method based on the Kimura 2-parameters model with 1,000 bootstrap replicates.

## Full genome sequencing of *Campylobacter jejuni* isolate

Full genome sequencing was performed in order to confirm taxonomic affiliation of the *C. jejuni* strain isolated from farm #3 samples. Briefly, isolated single colonies were grown overnight in brain heart infusion broth. Bacterial DNA was extracted using the PowerMag® Microbiome Kit (MoBio) following the manufacturer's instructions. The library was prepared with the Illumina Nextera XT v2 DNA Library Prep Kit and then sequenced on the Illumina NextSeq 500 system with 2 x 150 bp paired-end reads. Quality of the sequencing data was controlled with fastp v0.12.6 with default parameters [60]. Assembly was performed using MEGA-HIT assembler v1.1 [61] and reads were mapped on the contigs using Bowtie 2 [62] (default parameters with no secondary alignment) in order to obtain the mapping rate of each contig. This Whole Genome Shotgun project has been deposited at DDBJ/ENA/GenBank under the accession JACBXE000000000.

## Antibiosis activity of gut commensal species against *Campylobacter jejuni*

A selection of aerobic and anaerobic strains isolated from the caeca of high-performer chicken were chosen to cover the phylogenetic diversity observed in 16S rRNA repertoires. They were first screened for antibiosis activity against *Campylobacter jejuni* ATCC 700819. Commensal strains presenting activity against this reference strain were then tested against a *C. jejuni* strain isolated from a low-performer chicken. *Campylobacter* strains were cultivated on CHOCO + PVS agar plates (BioRad) at 42˚C in microaerophilic conditions. In preparation of screening for antibiosis activity, 26 aerobic and facultative anaerobic commensal strains (S4 Table) isolated from high-performer chickens were cultivated overnight at 37˚C in BHI broth in aerobic condition. Two μl of culture were then spotted onto LB plates and incubated at 37˚C or 42˚C for 24 h in anaerobic or aerobic conditions. Colonies were then inactivated with chloroform. For target bacteria preparation, few colonies of each target strain were added in PBS to obtain at 0.5 MacFarland suspension. One-hundredth dilution of the suspension was performed in BHI + 0.7% agar that was then deposited on the chloroform-inactivated colonies. Plates were incubated at 42˚C for 24 h in microaerophilic conditions. After incubation, the plates were analyzed for the presence of clear inhibition zones surrounding the inactivated colonies, indicating antagonistic activity against *C. jejuni*. Only the strains presenting activity against *C. jejuni* ATCC 700819 when cultivated at 37˚C in aerobic conditions were then tested using additional growth conditions. A total of 32 strictly anaerobic commensal strains isolated from high-performer chickens were processed the same way as aerobic/facultative aerobic strains except that they were cultivated for 24 h to 48 h at 37˚C in mGAM in anaerobic condition. Two μl of culture were then spotted onto mGAM plates, with all strains being tested for antibiosis activity after growth for 24 h at 37˚C, whereas only 18 of 32 were also tested after growth for 24 h at 42˚C (S5 Table). Inhibition diameters were measured and scored with the following parameters: (+) for inhibition diameters ≥ 2 mm, (+/-) for 0.1 to 1.9 mm, (-) when

no inhibition was observed. In order to check whether the inhibition zones could be due to acidification of the medium, the same cultures were spotted on the plates supplemented with 0.025% phenol red.

## Supporting information

**S1 Fig. Individual variation of the relative abundance of the 5 main bacterial phyla in the caecal contents of 7-days old chicks raised in high-performance (#1 and #2) and low-performance (#3) farms.** $n$ = 10 points for each farm.
(TIF)

**S2 Fig. Antibiosis activity of commensal strains against *C. jejuni* Farm #3 isolate.** Inhibition diameters were measured and scored with the following parameters: (+) for inhibition diameters $\geq$ 2 mm, (+/-) for 0.1 to 1.9 mm, (-) when no inhibition was observed. nd: no done due to the absence of growth in tested conditions.
(TIF)

**S3 Fig. Strategy used to isolate and identify commensal bacteria from chicken's caeca.** Caecal content was diluted in reduced PBS and then part of the suspension was directly seeded on mGAM with or without antibiotics and on LBS agar plates that were incubated anaerobically (mGAM) and/or aerobically (LBS) to allow recovery of a variety of anaerobes and of *Lactobacillus* species, respectively. Another part of the suspension was subjected to heat or to ethanol selection treatment and then seeded on LB or on mGAM complemented with sodium taurocholate and rumen fluid before incubation in aerobic or anaerobic conditions to allow recovery of sporulated *Bacillus* species or sporulated anaerobes, respectively.
(TIF)

**S1 Table. Family-level OTU relative abundancies observed in caeca collected from the 3 different farms and the average between the 3 farms.** Three low-abundant families under 0.1% (*Mogibacteriaceae*, *Dehalobacteriaceae* and *Staphylococcaceae*) were removed from the table. '-': not detected.
(XLSX)

**S2 Table. Identification results obtained after MALDI-TOF analysis.** Erythromycin (ERY), Ciprofloxacin (CIP), Gentamycin (GEN), Vancomycin (VAN), Sulfamethoxazole (SX) and Neomycin (NEO) were used in the media to select a diversity of species. Heat exposure (80°C for 20 min) and 70% ethanol exposure for 1 or 4h were used to select sporulated bacteria.
(XLSX)

**S3 Table. Identification results obtained after partial 16S rRNA gene sequencing of the isolates.** The capacity of closest described species to form spore was investigated in the literature and is reported in column L. Number of isolates corresponding to one specific taxonomic affiliation based on 16S rRNA gene analysis and MALDI-TOF spectra similarities are indicated in columns M and N.
(XLSX)

**S4 Table. Antibiosis activities against *C. jejuni* ATCC 700819 and Farm #3 isolate detected in facultative anaerobic commensal strains.** Inhibition diameters were measured and scored with the following parameters: (+) for inhibition diameters $\geq$ 2 mm, (+/-) for 0.1 to 1.9 mm, (-) when no inhibition was observed. nd: no done due to the absence of growth in tested conditions.
(XLSX)

**S5 Table. Antibiosis activities against *C. jejuni* ATCC 700819 detected in strictly anaerobic commensal strains.** Inhibition diameters were measured and scored with the following parameters: (+) for inhibition diameters ≥ 2 mm, (+/-) for 0.1 to 1.9 mm, (-) when no inhibition was observed. '-': experiment was not performed.
(XLSX)

## Acknowledgments

We thank Laure Sapey-Triomphe (Bioaster) for her contribution in generating the sequencing data and Adrien Villain (Bioaster) for his help in analyzing these data. We also thank Harro Timmerman (Boehringer Ingelheim) for helpful discussion when writing the manuscript.

## Author Contributions

**Conceptualization:** Christine Andreoni, Vincent Thomas.

**Data curation:** Aurore Duquenoy, Noémie Boucher, Lilia Boucinha.

**Formal analysis:** Aurore Duquenoy, Noémie Boucher, Lilia Boucinha.

**Funding acquisition:** Christine Andreoni, Vincent Thomas.

**Investigation:** Aurore Duquenoy, Maryne Ania.

**Methodology:** Aurore Duquenoy, Maryne Ania, Lilia Boucinha, Christine Andreoni.

**Project administration:** Aurore Duquenoy, Frédéric Reynier, Christine Andreoni, Vincent Thomas.

**Resources:** Christine Andreoni.

**Software:** Lilia Boucinha.

**Supervision:** Frédéric Reynier, Vincent Thomas.

**Validation:** Lilia Boucinha, Vincent Thomas.

**Visualization:** Noémie Boucher, Lilia Boucinha.

**Writing – original draft:** Aurore Duquenoy.

**Writing – review & editing:** Christine Andreoni, Vincent Thomas.

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
