## [Decision Letter · Decision Letter 0]

28 May 2020

PONE-D-20-13216

Caecal microbiota compositions from 7-day-old chicks reared in high-performance and low-performance industrial farms and systematic culturomics to select strains with anti-Campylobacter activity

PLOS ONE

Dear Dr. Thomas,

Thank you for submitting your manuscript to PLOS ONE. After careful consideration, we feel that it has merit but does not fully meet PLOS ONE’s publication criteria as it currently stands. Therefore, we invite you to submit a revised version of the manuscript that addresses the points raised during the review process.

In your revision you must respond specific to all of the comments of the two reviewers.  In particular, the comments about your culturomics methods are mandatory.  Further, it is vital that you emphasize the novelty of these data as Reviewer 1 suggests that these data are 'standard' for poultry microbiota studies.  Failure respond to these and all of the points made by the reviewers will result in the rejection of the revised manusvript.

We look forward to receiving your revised manuscript.

Kind regards,

Michael H. Kogut, Ph.D.

Academic Editor

PLOS ONE

2. We note that you are reporting an analysis of a microarray, next-generation sequencing, or deep sequencing data set. PLOS requires that authors comply with field-specific standards for preparation, recording, and deposition of data in repositories appropriate to their field. Please upload these data to a stable, public repository (such as ArrayExpress, Gene Expression Omnibus (GEO), DNA Data Bank of Japan (DDBJ), NCBI GenBank, NCBI Sequence Read Archive, or EMBL Nucleotide Sequence Database (ENA)). In your revised cover letter, please provide the relevant accession numbers that may be used to access these data. For a full list of recommended repositories, see http://journals.plos.org/plosone/s/data-availability#loc-omics or http://journals.plos.org/plosone/s/data-availability#loc-sequencing.

"This work has received, through BIOASTER investment, funding from the French Government through the Investissement d’Avenir program (Grant No. ANR-10-AIRT-03). This study was partially funded by Boehringer Ingelheim Animal Health. Dr Christine Andreoni, who is employed by Boehringer Ingelheim Animal Health, was involved in study design, decision to publish, and preparation of the manuscript."

We note that you received funding from a commercial source: Boehringer Ingelheim Animal Health

Reviewers' comments:

Reviewer's Responses to Questions

**Comments to the Author**

1. Is the manuscript technically sound, and do the data support the conclusions?

Reviewer #1: Yes

Reviewer #2: Partly

2. Has the statistical analysis been performed appropriately and rigorously? 

Reviewer #1: Yes

Reviewer #2: No

3. Have the authors made all data underlying the findings in their manuscript fully available?

Reviewer #1: Yes

Reviewer #2: Yes

4. Is the manuscript presented in an intelligible fashion and written in standard English?

Reviewer #1: Yes

Reviewer #2: Yes

5. Review Comments to the Author

Reviewer #1: This is standard manuscript on chicken gut microbiota which does not bring too much of new knowledge but I appreciate that authors did not attempt to overestimate their results. They also tried to combine 16S rRNA sequencing with culturomics. I may argue whether the performance is cause of differences in gut microbiota composition, or vice versa, whether gut microbiota composition in this particular study is the cause for differential performance. Or, whether both factors were mutually independent in this study and simply chickens in different farms have slightly different microbiota composition, and also different farms are of different productivity. In fact, with the design of this study, I would prefer the last possibility, i.e. that everything what was recorded in this study is random, not causal since the farm effect, local environment, humidity, feed storage, litter management, human personnel, ventilation, heating or air conditioning, all of this will be of much greater effect on chicken performance than composition of gut microbiota. However, this is just my opinion and authors may have different. I therefore suggest to check the following minor points.

line 49, 55 and throughout rest of the manuscript, you use term „microbiome“. Microbiome should be used to describe genes and genomes of the whole microbial community. However when you preferentially consider viable bacteria, you should use term „microbiota“. Please correct.

l. 81 and throughout rest of the manuscript, you use term „16S“. Please use at least „16S rRNA“. This is laboratory slang not suitable for official written document.

l.99, you introduce Farm 2 and 3 but reader does not know whether these represent well or poorly performing farms. Please specify this in this place.

l.122-124, this is the reason for my concerns. Faecalibacterium is nearly always considered as an indicator of well performing gut. And you report it in poorly performing chickens. And on the other hand, Proteus, E. coli or even Klebsiella, these genera are commonly enriched in poorly performing animals or humans – and you report them as markers of good performance. Let it be, do not correct the text but be careful, this is unusual.

l. 144, do not start sentences with numbers. Reword the sentence or use words, three hundreds and forty seven (not the best option in this case). Correct it not only in this position but also in the rest of the manuscript.

l. 235, delete „inter-individual“

l. 382, what do you mean by allowed 25% mismatch? That every fourth nucleotide could differ in forward and reverse sequence? This quite a lot.

Reviewer #2: The manuscript compares the microbiota of a small subsample of broiler chicks in two high- and one poor-performing barn using culture-independent and culture-dependent methods. Furthermore, antibiosis against Campylobacter jejuni was examined. This is an interesting approach, and the manuscript is generally well written.

Following are general and specific comments:

General Comments

1. Additional information needs to be provided on the three barns. Why is the performance in the third barn poor? Different source of chicks? Different barn composition? Different husbandry methods? Different stocking densities? What is the cause of the higher mortality? It seems strange that very specific mortality numbers (e.g. two decimal places) are provided for the first two barns, opposed to no decimal points for the poor-performing barn. Also, information on how the birds were chosen should be indicated. More specifically, how was it determined that the chicks chosen are representative of the flock?

2. The connection with C. jejuni needs to be strengthened with justification. In this regard, how was the Campylobacter strain isolated determined to be C. jejuni? Was subspecies characterization of the isolate conducted? Relying on relative frequency of ASVs for Campylobacteriaceae is insufficient. There are multiple species of Campylobacter (≥ 35) along with multiple species of Arcobacter (and other genera), many of which are present in the intestine of birds. At minimum, quantitative PCR for C. jejuni should have been completed.

3. Targeting C. jejuni in chicks needs to be justified. Extensive research has shown that outbreaks of C. jejuni in broiler flocks occurs late in the production cycle. Of the 1000’s of chicks that my team has sampled for C. jejuni, isolation of the bacterium from chicks is very infrequent. Although the mechanisms, are not well understood, presence of maternal antibodies has been suggested as a possible mechanism. The current study is presumably targeting colonization resistance as a mechanism, which is valid, but not handled comprehensively in the manuscript as currently presented.

4. I am uncertain why “bacteriocin” production, or lack thereof of the bacteria from birds from the poor-performing barn was not examined? In this regard, it would have been useful if the authors had included a specific study hypothesis(es) to clarify what was being tested, and what the primary goals/objectives of the study where. The hypothesis presented on L66 is very non-specific, but suggests that colonization resistance was enhanced in the “high-performing barns”. In the introduction, the authors present information on population of the chick intestines by bacteria in caeal digesta and feces (i.e. “Fecal Microbial Transplant”), but the study focuses on bacteriocins specifically (neglecting other mechanisms of colonization resistance). May be a broader scope would have been beneficial. Again, defining the hypothesis/goals of the study would have been beneficial.

5. How do the authors know that the antibiosis was due to bacteriocins and not from other anti-bacterial agents (e.g. secondary metabolites). To confirm bacteriocins, isn’t proof that antibiosis was due to a protein necessary (e.g. loss of action by protease treatment). Given the focus on bacteriocins, more in depth characterization of antibiosis is warranted.

6. The analysis of the culturomics aspect of the study could be strengthened. For example, the application of bioinformatics methodologies would have been beneficial (opposed to the qualitative manner in which the data is presented). To what degree was the diversity of bacteria recovered reflected of diversity of the community (i.e. determined by Illumina sequencing)? The authors should examine recent references in which bioinformatics is applied to culturomics data. The isolation methods applied (e.g. reliance on short-term duration “direct plating” using a relatively limited number of media) would not be expected to recover fastidious taxa. As mentioned above, inclusion of a specific hypothesis would be useful to provide the reader with the goal of the culturomics methods applied. If the goal was to recover diverse obligate anaerobic taxa, this the methods applied are insufficient.

7. The abstract is very generally written, and more specific information on the findings of the study should be included.

8. Some of the information presented in the discussion is beyond the scope of the study.

Specific Comments

L43. There is no coverage of Campylobacter jejuni in the introduction. See general comments above on the need to include a specific hypothesis(es) and objectives.

L88. Is “high performance” farms accurate. The statistics provided suggest that these two barns are comparable to the industry standard.

L82. See general comment above on the rationale for targeting the microbiota of 7-day-old chicks when C. jejuni does not typically colonize such young birds in production settings.

L93. Why are families italicised but not phyla?

L112. Given the current limitations of Illumina sequencing (e.g. targeting one or two variable regions), care should be exercised in resolving ASVs at the genus level of resolution. Typically, relative densities of taxa at the genus and species level should be conducted using an ancillary method such as quantitative PCR.

L127. Although I have not personally characterized the caeal microbiota of chicks, the Shannon’s index of diversity seems to be very low (presented in Table 1). In older birds (e.g. 20 days-old), a Shannon’s index in the range of 7 is the norm. This should be handled in the discussion.

L130. The higher richness and alpha diversity of the caeal microbiota of chicks from the poor-performing barn is unexpected. This should be handled in the discussion.

L135. Inclusion of pair-wise Permanova results should be included (i.e. quantitative evidence for a difference among the three farms).

L141. Why only one caecum per farm for culturomics? Why weren’t bacteria recovered from birds on the poor-performing farm (see general comment)?

L147. Table 2 is of limited value. The value of including “primary culture” versus “sub-culture” should be clarified.

L162. See general comment on quantitative coverage of the culturomics information.

L184. Typo (“Bust”).

L196. See general comment on validation of bacteriocins.

L245. Quantitative PCR for Faecalibacterium and Campylobacter species is necessary to confirm this.

L247. This statement (i.e. “usually reported”) is not consistent with the epidemiological data of Campylobacter jejuni outbreaks in broiler barns. See general comment above. This aspect of the discussion must be expanded.

L253. This is beyond the scope of the study.

L289. See general comment on bacteriocins.

L334. See general comment on including additional information on the farms.

L353. How were the ten chicks chosen to ensure they were representative of the flock? Was the poor-performing barn consistently so? What was responsible for such high mortality? Was the mortality rates presented in Table 4, 7-day chick mortality, or overall mortality?

L381. Additional analyses? Eveness? Permanova? Line 407. Bacilli are class of Firmicutes. Separate treatment of Firmicutes (L411) is confusing.

L409. Technically, “dishes” not plates (i.e. Petri dishes).

L420. Identification of Campylobacter to species? How done? Other characterization? Strain diversity? See general comment.

L445. Near full length

L458. How do you know bacterocins, and not other antibacterial compounds. See L473 “putative”? See also general comment above on validation of bacteriocin activity.

L464. Why only 26 strains? L474 Why only 32 obligate anaerobes?

L465. Why only high-performer chickens?

L683. Figure 1 is of limited usefulness. Demote to supplemental.

L691. Figure 3 is of limited usefulness given its resolution. There are 35+ species of Campylobacter, multiple species of Arcobacter, and other genera, many of which are common in chickens. Quantitative PCR would have been preferable. See general comment.

L693. See general comment regarding resolution of Illumina sequence analysis.

L697. As mentioned by the authors, there is extensive inter-bird variation in richness. That the richness is highest at Farm 3 is an interesting observation, and this should be discussed in the discussion section.

L700. Why did you choose to use a Bray-Curtis dissimilarity metric? Opposed to UniFrac (differs from dissimilarity measures such as Bray-Curtis dissimilarity in that it incorporates information on the relative relatedness of community members by incorporating phylogenetic distances between observed organisms in the computation)? Statistical analyses among farms should be completed (e.g. pairwise Permanova analysis).

L702. Figure 7 was very low resolution and not possible to evaluate as a result.

Although generally well written, there are some minor grammatical issues throughout the manuscript that should be corrected.

6. PLOS authors have the option to publish the peer review history of their article (what does this mean?). If published, this will include your full peer review and any attached files.

Reviewer #1: No

Reviewer #2: No

---

## [Author Response · Author response to Decision Letter 0]

10 Jul 2020

We ensured that the manuscrpit meets PLOS ONE's style requirements.

All sequencing data has been deposited in GenBank and SRA and accession numbers were provided in the revised version of the manuscript.

The amended Competing Interests Statement is provided in the Revised Cover Letter.

The mentions 'data not shown' were removed from the manuscript.

Reviewer #1: 

Response to Reviewer 1's general comment:

We do agree with Reviewer 1 that it is difficult to draw any strong conclusion from our study. However we still believe the study brings a significant amount of novelty compared to already published studies, and that this information could be useful for researchers in the field.

The initial goal of the study was to cultivate a wide diversity of bacterial commensal species from chicks collected from farms presenting good performances, and then to investigate the new isolates for specific properties that might be of interest to build cocktails of strains that could be administered to very young chicks or even in ovo before birth. In this perspective, we were also interested to investigate which bacteria (OTUs) could be differentially present in high (HP) vs low performance (LP) young chicks, in order to focus on species (genera) that are more abundant in 7-days old HP and that might ‘shape’ adult chicks microbiota (1, 2). That’s why we choose to sequence the 16S rRNA gene repertoires on the same samples that were collected for culturomics experiments, and that’s also why we choose not to do culturomics with samples collected from LP, with the assumption that bacteria collected from these animals would not be the best possible candidates to build cocktails of beneficial strains. Due to the design of the experiments, the sequencing results were obtained while the cultivation experiments were already underway or even completed. We were very intrigued by the overabundance of Campylobacter-affiliated OTUs in the LP group and since there is a significant amount of literature that tends to demonstrate that C.jejuni is rather detrimental we choose to investigate the isolates collected from HP animals for antibiosis activity against C.jejuni. We could not do the same with LP since these samples were not cultivated, which we agree constitutes a bias in the study. In addition, other investigations are underway to characterize potential probiotic candidates isolated from HP birds but these results are still confidential and will be included in a separate report.

We believe that the main novel points of importance are:

- The variety of culture conditions used, with or without ethanol or heat preselection treatment, and the use of rumen fluid as a nutrients source and sodium taurocholate as a spore germinant. There are not so many ‘culturomics’ studies that were applied to chicken microbiota and, to our knowledge, ours is the first one that used these culture conditions. Importantly, we believe that the use of ethanol selection followed by seeding on Gifu Anaerobic Medium supplemented with rumen fluid and sodium taurocholate is an interesting option that gives access to a large diversity of strictly anaerobic, sporulated commensals. Recently published culturomics studies came to similar conclusions (3). 

- The fact that we were able to cultivate C. jejuni from samples collected from very young chicks, which is unusual. In order to answer to Reviewer 2’s concerns we sequenced the full genome of the C.jejuni isolate and were able to confirm species identity (see reply to Reviewer 2).

- The fact that antibiosis activity against both C. jejuni isolates is only found in already known (mainly Lactic Acid) bacteria, whereas none of the strictly anaerobic bacteria tested in our study showed activity against these two isolates. Several recent studies reported bacteriocin activity of human commensal gut species (including Blautia species) against Enterococcus, Prevotella or pathogenic Clostridiales (4-6). The fact that we did not find any antibiosis activity of our strictly anaerobic strains against C. jejuni tends to demonstrate that this kind of activity is probably not frequent for strictly anaerobic commensal species. 

- The fact that antibiosis activity against both C. jejuni isolates can vary significantly according to conditions used to cultivate bacteria. Antibiosis, mainly due to bacteriocin production by various commensal species, has been largely reported in the literature but, at least to our knowledge, variation in antibiosis activity depending on temperature and oxygen conditions is a major point that has not been reported so far. Since eggs are incubated at approx. 37°C in hatcheries whereas the internal temperature of adult chickens is 42°C, and since the digestive tract of young chicks is aerobic before turning anaerobic after a few days, this parameter should be taken into account when designing cocktails of strains intended to be used for in ovo or in vivo administration. To our opinion, this should be brought to the knowledge of researchers who are exploring commensal strains for antibiosis (bacteriocin) activity against C.jejuni (7-10) or against other pathogenic species. 

Answers to Reviewer 1's specific comments:

• line 49, 55 and throughout rest of the manuscript, you use term „microbiome“. Microbiome should be used to describe genes and genomes of the whole microbial community. However when you preferentially consider viable bacteria, you should use term „microbiota“. Please correct.

This has been corrected.

• l. 81 and throughout rest of the manuscript, you use term „16S“. Please use at least „16S rRNA“. This is laboratory slang not suitable for official written document.

This has been corrected.

• l.99, you introduce Farm 2 and 3 but reader does not know whether these represent well or poorly performing farms. Please specify this in this place.

This has been added.

• l.122-124, this is the reason for my concerns. Faecalibacterium is nearly always considered as an indicator of well performing gut. And you report it in poorly performing chickens. And on the other hand, Proteus, E. coli or even Klebsiella, these genera are commonly enriched in poorly performing animals or humans – and you report them as markers of good performance. Let it be, do not correct the text but be careful, this is unusual.

We agree with Reviewer 1 that these results are quiet unusual, and that one would rather expect to observe higher abundancy of Faecalibacterium in HP than in LP chicks and higher abundance of E. coli in LP than in HP chicks. However, similar unusual observations were already reported by others for both F. prausnitzii (11) and E. coli (12). In human, what is usually termed ‘F. prausnitzii’ likely corresponds to a variety of phylogroups and even species (13). Whether the same is true for poultry, and if all putative Faecalibacterium phylogroups/species/strains can be associated with good feeding performance needs to be further explored. This should be done with sequencing technologies offering higher phylogenetic resolution than the classical V3-V4 sequencing strategy that was used in our study. Concerning enterobacteria, it is well reported that they are early colonizers of the chick’s digestive tract and that they can still be very abundant in Day-7 chicks (14).Taken together, this contradictory information underlines the fact that microbiota-performance association studies are still very preliminary and should therefore not be taken for granted.

• l. 144, do not start sentences with numbers. Reword the sentence or use words, three hundreds and forty seven (not the best option in this case). Correct it not only in this position but also in the rest of the manuscript.

Thank you for pointing this. It has been corrected in the manuscript.

• l. 235, delete „inter-individual“

Done

• l. 382, what do you mean by allowed 25% mismatch? That every fourth nucleotide could differ in forward and reverse sequence? This quite a lot.

We used the default values of the FLASH software, that has been demonstrated to be the optimal value in the original paper (15):

Figure taken from the original article: Impact of the mismatch ratio parameter on correctness of the readmerging algorithm. The horizontal axis shows the number of incorrectlymerged read pairs, and the vertical axis shows the number of correctly mergedread pairs. The mismatch ratio parameter is shown at each point along thegraph

Response to Reviewer 2's general comment 1:

We have collected additional information concerning the three barns, it has been added in the M&M section and in Table 4. The source of chicks was the same for farms 2 and 3, whereas it was different for farm 1. Stocking density was identical for farms 1 and 2 whereas chicks were raised in free-range for farm 3. Water was treated with peroxide hydrogen for farm 1, with chlorine dioxide for farm 2, and was left untreated for farm 3. Birds did not receive any antibiotic treatment before sampling for the three farms.

The main new information is that farm 3 is an organic farm in which chicks were raised in free-range. This could likely contribute to explain higher microbiota diversity observed in these animals as well as early colonization by C.jejuni and higher overall mortality. We did not get this information when the first version of the manuscript was submitted. 

Overall mortality is indicated in Table 4. The two decimal places issue is due to Excel that automatically deleted the two ‘0’ after the decimal point. This has been corrected. 

Birds were randomly selected, so we cannot add information on any specific process used for choosing the birds. Our initial hypothesis was that there should be less difference in microbiota composition between chicks collected from the same farm than in microbiota composition between chicks collected from different farms. Therefore we did not determine chick’s representativity before sampling. However it seems that the initial hypothesis was relatively correct according to phylogenetic clustering of the 30 samples represented in revised Figure 1. 

Response to Reviewer 2's general comment 2:

We do agree with Reviewer 2 that it is not possible to rely on partial 16S rRNA encoding gene sequencing to identify C. jejuni at the species level. However we were able to isolate and cultivate Campylobacter species from these samples using the RAPID'Campylobacter agar that is selective for C. jejuni, C. coli and C. lari. In order to obtain a more precise identification and to answer to Reviewer 2’s concern, the full genome of the isolate has been sequenced and deposited on GenBank (accession number JACBXE000000000). Based on this additional information we can confirm that the isolate belongs to the C. jejuni species (97.59% average nucleotide identity value between the genome sequence and C. jejuni reference genome GCF_000009085.1).

Response to Reviewer 2's general comment 3:

We do understand Reviewer 2’s concern. Actually, we are not specialists of chicken microbiota and even less specialists of Campylobacter colonization. Comments/suggestions made by researchers with much more experience are therefore greatly appreciated. As explained in our reply to Reviewer 1 our initial goal was not to target Campylobacter and look for antibiosis activity against this species, but rather to mine microbiota collected from High Performing (HP) birds for potential growth-promoting strains. The detection followed by cultivation of Campylobacter from the Low Performing (LP) birds was incidental. Since we had built a collection of commensal strains from the HP birds we choose to explore possible antibiosis activity against Campylobacter as a first criteria to identify probiotic candidate in our collection. This is one aspect but we are currently exploring many other phenotypic traits that are still confidential and will be reported in further publications. We’ve added several sentences in the introduction to better introduce the colonization resistance hypothesis. 

Response to Reviewer 2's general comment 4:

We do agree with this comment. However, as already mentioned above we did not cultivate commensal bacteria from LP birds, since the initial strategy was to build a collection of strains isolated from HP birds. We have modified the introduction to better explain the course of the study. We hope it will be acceptable for Reviewer 2.

Response to Reviewer 2's general comment 5:

Even if bacteriocin production by species identified as bacteriocin producers in our study has been frequently reported (8, 16-20), we do agree with Reviewer 2 that it could possibly be due to other anti-bacterial agents. We did check that antibiosis was not due to acidification. Unfortunately we did not have the opportunity to better characterize antibiosis for this report but we hope we can spend more effort on this characterization in the near future. We modified the text of the manuscript to better reflect that antibiosis could be due to non-bacteriocin compounds and that additional experiments are needed for a better understanding. We hope this will be acceptable for Reviewer 2. 

Response to Reviewer 2's general comment 6:

Additional information was added in the manuscript to answer Reviewer 2’s comment. We calculated that 8.5% of the 6 105 OTUs retrieved from 16S repertoire analysis could be cultivated. These cultivated OTUs accounted for 49.2% abundance of the total OTUs. At the family level, the only taxon for which OTUs were detected in significant amount (> 1%) but for which representative isolates were not cultivated was the Bacteroidaceae family. Figure 6 has been added in the manuscript to highlight this information. This is likely due to the fact that OTUs affiliated to this family (all affiliated to the Bacteriodes genus) were detected in significant amounts only in cecal samples collected from farm #1, and that they were not equally distributed, with 4 samples presenting relative abundancies >10% and 5 with relative abundancies <1%, the last one being at 3.1%. Farm#1 sample #2 that was randomly chosen for cultivation without ethanol pre-treatment contained only 0.1% of OTUs affiliated to the Bacteriodes genus, thus explaining why we were not able to recover members of this family in culture. This was not an important issue for us since in the perspective of developing probiotic candidates we were mainly interested in cultivating aerobic and facultative anaerobic species, or strictly anaerobic species that form spores that can resist oxygen exposure. Bacteroides species are strictly anaerobic and they do not form spores, so they are unlikely to be used as probiotics.

This information has been added in the manuscript and we modified revised figures 1 and 2 to highlight which samples were used for cultivation with or without ethanol selection treatment. 

Response to Reviewer 2's general comment 7:

We extensively modified the abstract to give more information on the findings of the study.

Response to Reviewer 2's general comment 8:

See answer to specific comment L.253

Answers to Reviewer 2's specific comments

L43. There is no coverage of Campylobacter jejuni in the introduction. See general comments above on the need to include a specific hypothesis(es) and objectives.

Hypothesis and objectives have been included in the introduction, as well as coverage of C. jejuni. 

L88. Is “high performance” farms accurate. The statistics provided suggest that these two barns are comparable to the industry standard.

The term ‘high performance’ was used to reflect the fact that farms n°& and 2 perform better than farm 3. Performance ranking was calculated based on average body weight, average daily weight gain and overall mortality. 

L82. See general comment above on the rationale for targeting the microbiota of 7-day-old chicks when C. jejuni does not typically colonize such young birds in production settings.

See above. Our initial goal was not really to target activity against C.jejuni, but rather to build a collection of commensal strains that can then be investigated for probiotic candidates. The finding and isolation of C. jejuni was incidental and we therefore chose to test antagonistic activity against this zoonotic species.

L93. Why are families italicised but not phyla?

Genera and species names should always be in italics whereas conventions vary for higher taxonomic ranks. See open access paper: https://onlinelibrary.wiley.com/doi/abs/10.1002/9781118960608.bm00004

We are happy to write phyla’s names in italics if required by PLOS One Editors.

L112. Given the current limitations of Illumina sequencing (e.g. targeting one or two variable regions), care should be exercised in resolving ASVs at the genus level of resolution. Typically, relative densities of taxa at the genus and species level should be conducted using an ancillary method such as quantitative PCR.

We do agree with this remark. V3-V4 sequencing gives only limited taxonomic information and genera lists identified from these sequencing data should be interpreted with caution. However we used classical 16S annotation pipelines that are widely used in the scientific community and for which advantages and limitations are known and described. We were able to identify only one colony morphology when cultivating frozen LP caecal samples on RAPID’Campylobacter Agar. We confirmed that these bacteria belong to C. jejuni species by performing full genome sequencing (see above).

L127. Although I have not personally characterized the caeal microbiota of chicks, the Shannon’s index of diversity seems to be very low (presented in Table 1). In older birds (e.g. 20 days-old), a Shannon’s index in the range of 7 is the norm. This should be handled in the discussion.

We have added a sentence to discuss this point.

L130. The higher richness and alpha diversity of the caeal microbiota of chicks from the poor-performing barn is unexpected. This should be handled in the discussion.

We now discuss this point in light of the fact that the poor-performing farm was a free-range farm, which can explain why higher richness and alpha diversity were observed in these animals.

L135. Inclusion of pair-wise Permanova results should be included (i.e. quantitative evidence for a difference among the three farms).

As suggested by Reviewer 2, a Permanova test was performed under the null hypothesis (H0) that the mean and the dispersion of both groups are equivalent. This test is an analysis of variance method based on dissimilarities and non-parametric significance. A P-value for H0 is computed by comparing the fisher Statistic to a permutated Fisher statistic (over 1000 permutation). This test discriminate groups based on one or more factor (explicative variables). In this work, performance between farms (high/low) was used as the only factor. A significant difference was found between high and low performance farms with a p-value of 0.001. This result has been added in the manuscript.

L141. Why only one caecum per farm for culturomics? Why weren’t bacteria recovered from birds on the poor-performing farm (see general comment)?

We agree with Reviewer 2 that cultivating only one caecum per farm is an important limitation in our study. From a very practical point of view, the reason for such a limited amount of cultured caeca is that we were not able to handle more than one sample per farm at a time due to the variety of growth conditions used, then followed by colony screening, sub-culture and MALDI-TOF analysis (347 isolates were obtained from farm 1, 234 from farm 2). In retrospect, it would probably have been more appropriate to pool the samples prior to conducting the culture experiments. However, and as described in new Figure 6, we were still able to recover a significant proportion of species corresponding to the most abundant OTUs identified after 16S repertoire analysis. As already mentioned in answer to point 6, the most notable exception was for members of the Bacterioides genus that were un-uniformly distributed between chicks from farm #1, and for which the sample used for cultivation contained very few (0.1%) OTUs affiliated to this genus whereas several other samples contained considerable amounts (Fig.1 and Fig.2). 

We did not try to cultivate additional isolates from frozen caecal samples since freezing likely affects viability of selected species. As explained above, we did not culture bacteria from the poor-performing farm since culture was conducted to establish a collection of commensal species that could then be mined for probiotic candidates, which was intellectually easier to conceive with isolates collected from high- than from low-performing farms. However we cannot exclude that good probiotic candidates could also possibly be isolated from low-performing farms.

L147. Table 2 is of limited value. The value of including “primary culture” versus “sub-culture” should be clarified.

We believe table 2 gives a good overview of the number of isolates that were recovered using different culture conditions. The value of including “primary culture” versus “sub-culture” is that it highlights the fact that an important number of isolates were lost during the culture process, which is an important information.

L162. See general comment on quantitative coverage of the culturomics information.

See reply to point 6. We hope this will be acceptable for Reviewer 2.

L184. Typo (“Bust”).

Corrected

L196. See general comment on validation of bacteriocins.

See our answer in the general comment section.

L245. Quantitative PCR for Faecalibacterium and Campylobacter species is necessary to confirm this.

We agree with Reviewer 2 that qPCR would be an interesting additional approach. However, to our knowledge, the vast majority of 16S-based compositional analysis studies are usually not confirmed by qPCR experiments.

L247. This statement (i.e. “usually reported”) is not consistent with the epidemiological data of Campylobacter jejuni outbreaks in broiler barns. See general comment above. This aspect of the discussion must be expanded.

We agree with Reviewer 2 that this part of the discussion was confusing. We did not have the information that the low-performing birds were collected from a free-range farm when the first version of the manuscript was submitted. This is an important point that was taken into consideration in the revised version of the manuscript. It has been reported that C. jejuni colonization can potentially appear at 2 weeks of age in birds raised in these farms (21), which is confirmed in our study. 

L253. This is beyond the scope of the study.

We do not completely agree with this point. It seems that butyrate-sensing is an important factor for C. jejuni colonization, as reported in several recent papers (22-24). We therefore believe that the fact that OTUs related to C. jejuni were detected in animals in which OTUs of the well-known butyrate producer F. prausnitzii were also detected in more important abundance should be discussed in our manuscript. 

Of note, both butyrate-sensing genes reported in the recent paper published by Goodman et al (23) were detected in the genome of our C. jejuni isolate.

L289. See general comment on bacteriocins.

See our reply. We hope this will be acceptable for Reviewer 2.

L334. See general comment on including additional information on the farms.

Done

L353. How were the ten chicks chosen to ensure they were representative of the flock? Was the poor-performing barn consistently so? What was responsible for such high mortality? Was the mortality rates presented in Table 4, 7-day chick mortality, or overall mortality?

See reply to general comment 1.

Concerning mortality, unfortunately we were not able to obtain more information on the cause of high mortality observed in farm 3. Overall mortality is indicated in Table 4.

L381. Additional analyses? Eveness? Permanova? Line 407. Bacilli are class of Firmicutes. Separate treatment of Firmicutes (L411) is confusing.

See comment on L135: a Permanova test was performed. We modified the distinction between Bacilli and Firmicutes in this part of the manuscript, and also modified Table 2 to indicate culture conditions rather than targeted species.

L409. Technically, “dishes” not plates (i.e. Petri dishes).

Even if we agree that the term ‘dishes’ could be more appropriate, it seems to us that the term ‘plates’ is widely used. We are happy to change wording if required. 

L420. Identification of Campylobacter to species? How done? Other characterization? Strain diversity? See general comment.

We chose to sequence the genome of one C. jejuni isolate to confirm identification. It does not cover potential strain diversity but we hope this level of information will be acceptable for Reviewer 2.

L445. Near full length

Corrected

L458. How do you know bacterocins, and not other antibacterial compounds. See L473 “putative”? See also general comment above on validation of bacteriocin activity.

In fact, as mentioned in reply to comment 5 we cannot be sure that these are bacteriocins and not other antibacterial compounds. That is why we used the term ‘putative’ in the first version of the manuscript. See reply to comment 5 for more detailed answer to this point.

L464. Why only 26 strains? L474 Why only 32 obligate anaerobes?

There are several reasons for this. Some of the isolates among aerobic / facultative aerobic strains developed very quickly, resulting in complete invasion of the plates, thus rendering the test difficult to perform. This was the case for Proteus strains for instance. Several Bacillus species resisted chloroform inactivation, likely due to the formation of spores, and then invaded the plates during incubation for C. jejuni growth. Some of the strictly anaerobic strains formed only very little colonies or even no colonies since we chose to perform these tests in the absence of rumen fluid in the media to avoid any possible inhibition of C. jejuni growth by un-identified compounds potentially present in rumen fluid. Overall, the goal of this experiment was to investigate if potential production of antibiosis against C. jejuni was a common feature in commensal species, which did not necessitate to test all available strains to answer.

L465. Why only high-performer chickens?

See reply to general comment 3. 

L683. Figure 1 is of limited usefulness. Demote to supplemental.

Done

L691. Figure 3 is of limited usefulness given its resolution. There are 35+ species of Campylobacter, multiple species of Arcobacter, and other genera, many of which are common in chickens. Quantitative PCR would have been preferable. See general comment.

We agree that this figure is of limited usefulness. It has been removed from the revised manuscript.

L693. See general comment regarding resolution of Illumina sequence analysis.

We agree with Reviewer 2 that the phylogenetic resolution of V3-V4 sequencing using Illumina (or any other) technology is relatively limited, and that it might be a little bit overoptimistic to assign these short sequences to genera rather than to families. However we had the feeling that this is the classical way this kind of data is analyzed, and that we should therefore process our data in a similar way. 

L697. As mentioned by the authors, there is extensive inter-bird variation in richness. That the richness is highest at Farm 3 is an interesting observation, and this should be discussed in the discussion section.

This is likely due to the fact that Farm 3 is an organic farm in which birds have access to free range, which might result in higher bacterial richness in the chick’s environment and therefore higher microbiota richness. This has been added in the discussion.

L700. Why did you choose to use a Bray-Curtis dissimilarity metric? Opposed to UniFrac (differs from dissimilarity measures such as Bray-Curtis dissimilarity in that it incorporates information on the relative relatedness of community members by incorporating phylogenetic distances between observed organisms in the computation)? Statistical analyses among farms should be completed (e.g. pairwise Permanova analysis).

We thank the reviewer for this suggestion. We have also used UniFrac as a dissimilarity metric, and the resulting PCoA did not display much difference between farms, contrary to that obtained with the Bray Curtis dissimilarity metric. We hypothesized that the discriminating taxa between high and low-performance farms were actually related (e.g. belonged to the same families), which resulted in a low UniFrac distance. However, we know that within a given taxon (e.g. family, genus and even species), the functions encoded in the genomes of the different strains may differ. We felt that the UniFrac metrics, while actually very powerful to differentiate samples from diverse environments, may in this case play down real differences between farms with low and high performances. 

A Permanova test was also performed under the null hypothesis (H0) that the mean and the dispersion of both groups are equivalent. Performance between farms (high/low) was used as the only factor. A significant difference was found between high and low performance farms with a p-value of 0.001. 

L702. Figure 7 was very low resolution and not possible to evaluate as a result.

We apologize for the inconvenience, but it appears that the pdf files generated from the files uploaded during the submission process are of limited resolution. 

Although generally well written, there are some minor grammatical issues throughout the manuscript that should be corrected.

We have tried to correct any grammatical issue, we hope the manuscript will be acceptable in this version.

REFERENCES

1. Wilson KM, Rodrigues DR, Briggs WN, Duff AF, Chasser KM, Bielke LR. Evaluation of the impact of in ovo administered bacteria on microbiome of chicks through 10 days of age. Poult Sci. 2019.

2. Rodrigues DR, Winson E, Wilson KM, Briggs WN, Duff AF, Chasser KM, et al. Intestinal Pioneer Colonizers as Drivers of Ileal Microbial Composition and Diversity of Broiler Chickens. Front Microbiol. 2019;10:2858.

3. Diakite A, Dubourg G, Dione N, Afouda P, Bellali S, Ngom, II, et al. Optimization and standardization of the culturomics technique for human microbiome exploration. Sci Rep. 2020;10(1):9674.

4. Chiumento S, Roblin C, Kieffer-Jaquinod S, Tachon S, Lepretre C, Basset C, et al. Ruminococcin C, a promising antibiotic produced by a human gut symbiont. Sci Adv. 2019;5(9):eaaw9969.

5. Coyne MJ, Bechon N, Matano LM, McEneany VL, Chatzidaki-Livanis M, Comstock LE. A family of anti-Bacteroidales peptide toxins wide-spread in the human gut microbiota. Nat Commun. 2019;10(1):3460.

6. Kim SG, Becattini S, Moody TU, Shliaha PV, Littmann ER, Seok R, et al. Microbiota-derived lantibiotic restores resistance against vancomycin-resistant Enterococcus. Nature. 2019;572(7771):665-9.

7. Messaoudi S, Kergourlay G, Dalgalarrondo M, Choiset Y, Ferchichi M, Prevost H, et al. Purification and characterization of a new bacteriocin active against Campylobacter produced by Lactobacillus salivarius SMXD51. Food Microbiol. 2012;32(1):129-34.

8. Robyn J, Rasschaert G, Messens W, Pasmans F, Heyndrickx M. Screening for lactic acid bacteria capable of inhibiting Campylobacter jejuni in in vitro simulations of the broiler chicken caecal environment. Benef Microbes. 2012;3(4):299-308.

9. Scerbova J, Laukova A. Sensitivity to Enterocins of Thermophilic Campylobacter spp. from Different Poultry Species. Foodborne Pathog Dis. 2016;13(12):668-73.

10. Johnson TJ, Shank JM, Johnson JG. Current and Potential Treatments for Reducing Campylobacter Colonization in Animal Hosts and Disease in Humans. Front Microbiol. 2017;8:487.

11. Yan W, Sun C, Yuan J, Yang N. Gut metagenomic analysis reveals prominent roles of Lactobacillus and cecal microbiota in chicken feed efficiency. Sci Rep. 2017;7:45308.

12. Torok VA, Hughes RJ, Mikkelsen LL, Perez-Maldonado R, Balding K, MacAlpine R, et al. Identification and characterization of potential performance-related gut microbiotas in broiler chickens across various feeding trials. Appl Environ Microbiol. 2011;77(17):5868-78.

13. Fitzgerald CB, Shkoporov AN, Sutton TDS, Chaplin AV, Velayudhan V, Ross RP, et al. Comparative analysis of Faecalibacterium prausnitzii genomes shows a high level of genome plasticity and warrants separation into new species-level taxa. BMC Genomics. 2018;19(1):931.

14. Ballou AL, Ali RA, Mendoza MA, Ellis JC, Hassan HM, Croom WJ, et al. Development of the Chick Microbiome: How Early Exposure Influences Future Microbial Diversity. Front Vet Sci. 2016;3:2.

15. Magoc T, Salzberg SL. FLASH: fast length adjustment of short reads to improve genome assemblies. Bioinformatics. 2011;27(21):2957-63.

16. Svetoch EA, Eruslanov BV, Levchuk VP, Perelygin VV, Mitsevich EV, Mitsevich IP, et al. Isolation of Lactobacillus salivarius 1077 (NRRL B-50053) and characterization of its bacteriocin, including the antimicrobial activity spectrum. Appl Environ Microbiol. 2011;77(8):2749-54.

17. Messaoudi S, Kergourlay G, Rossero A, Ferchichi M, Prevost H, Drider D, et al. Identification of lactobacilli residing in chicken ceca with antagonism against Campylobacter. Int Microbiol. 2011;14(2):103-10.

18. Stern NJ, Svetoch EA, Eruslanov BV, Perelygin VV, Mitsevich EV, Mitsevich IP, et al. Isolation of a Lactobacillus salivarius strain and purification of its bacteriocin, which is inhibitory to Campylobacter jejuni in the chicken gastrointestinal system. Antimicrob Agents Chemother. 2006;50(9):3111-6.

19. Svetoch EA, Stern NJ, Eruslanov BV, Kovalev YN, Volodina LI, Perelygin VV, et al. Isolation of Bacillus circulans and Paenibacillus polymyxa strains inhibitory to Campylobacter jejuni and characterization of associated bacteriocins. J Food Prot. 2005;68(1):11-7.

20. Cole K, Farnell MB, Donoghue AM, Stern NJ, Svetoch EA, Eruslanov BN, et al. Bacteriocins reduce Campylobacter colonization and alter gut morphology in turkey poults. Poult Sci. 2006;85(9):1570-5.

21. Ijaz UZ, Sivaloganathan L, McKenna A, Richmond A, Kelly C, Linton M, et al. Comprehensive Longitudinal Microbiome Analysis of the Chicken Cecum Reveals a Shift From Competitive to Environmental Drivers and a Window of Opportunity for Campylobacter. Front Microbiol. 2018;9:2452.

22. Luethy PM, Huynh S, Ribardo DA, Winter SE, Parker CT, Hendrixson DR. Microbiota-Derived Short-Chain Fatty Acids Modulate Expression of Campylobacter jejuni Determinants Required for Commensalism and Virulence. mBio. 2017;8(3).

23. Goodman KN, Powers MJ, Crofts AA, Trent MS, Hendrixson DR. Campylobacter jejuni BumSR directs a response to butyrate via sensor phosphatase activity to impact transcription and colonization. Proc Natl Acad Sci U S A. 2020;117(21):11715-26.

24. Hankel J, Jung K, Kuder H, Keller B, Keller C, Galvez E, et al. Caecal Microbiota of Experimentally Campylobacter jejuni-Infected Chickens at Different Ages. Front Microbiol. 2019;10:2303.

---

## [Editor Report · Decision Letter 1]

29 Jul 2020

Caecal microbiota compositions from 7-day-old chicks reared in high-performance and low-performance industrial farms and systematic culturomics to select strains with anti-Campylobacter activity

PONE-D-20-13216R1

Dear Dr. Thomas,

We’re pleased to inform you that your manuscript has been judged scientifically suitable for publication and will be formally accepted for publication once it meets all outstanding technical requirements.

Kind regards,

Michael H. Kogut, Ph.D.

Academic Editor

PLOS ONE
---

## [Editor Report · Acceptance letter]

12 Aug 2020

PONE-D-20-13216R1 

Caecal microbiota compositions from 7-day-old chicks reared in high-performance and low-performance industrial farms and systematic culturomics to select strains with anti-Campylobacter activity 

Dear Dr. Thomas:

I'm pleased to inform you that your manuscript has been deemed suitable for publication in PLOS ONE. Congratulations! Your manuscript is now with our production department. 

Kind regards, 

on behalf of

Dr. Michael H. Kogut 

Academic Editor

PLOS ONE